# Excessive serine from the bone marrow microenvironment impairs megakaryopoiesis and thrombopoiesis in Multiple Myeloma

Chunmei Kuang[1,2,7], Meijuan Xia[3,7], Gang An[3], CuiCui Liu[3], Cong Hu[1,2], Jingyu Zhang[1,2], Zhenhao Liu [1,2], Bin Meng[1,2], Pei Su[3], Jiliang Xia[1,2], Jiaojiao Guo[1,2], Yinghong Zhu[1,2], Xing Liu[1,2], Xuan Wu[1,2], Yi Shen[4], Xiangling Feng[5], Yanjuan He[1], Jian Li [6], Lugui Qiu[3], Jiaxi Zhou [3] ✉ & Wen Zhou [1,2] ✉

Thrombocytopenia is a major complication in a subset of patients with multiple myeloma (MM). However, little is known about its development and significance during MM. Here, we show thrombocytopenia is linked to poor prognosis in MM. In addition, we identify serine, which is released from MM cells into the bone marrow microenvironment, as a key metabolic factor that suppresses megakaryopoiesis and thrombopoiesis. The impact of excessive serine on thrombocytopenia is mainly mediated through the suppression of megakaryocyte (MK) differentiation. Extrinsic serine is transported into MKs through SLC38A1 and downregulates SVIL via SAM-mediated tri-methylation of H3K9, ultimately leading to the impairment of megakaryopoiesis. Inhibition of serine utilization or treatment with TPO enhances megakaryopoiesis and thrombopoiesis and suppresses MM progression. Together, we identify serine as a key metabolic regulator of thrombocytopenia, unveil molecular mechanisms governing MM progression, and provide potential therapeutic strategies for treating MM patients by targeting thrombocytopenia.

Multiple myeloma (MM) is the second most common hematologic malignancy with high incidence and mortality rates[1]. Clonal proliferation of malignant plasma cells within the BMME results in a host of clinical manifestations including hypercalcemia, renal failure, lytic bone lesions, and anemia, infection and bleeding induced by cytopenia[2]. Thrombocytopenia has also been observed in patients with MM, although it is a relatively less common complication[3–6]. The incidence of thrombocytopenia and its significance in the development of MM are still poorly defined.

In the bone marrow (BM), the cells of the megakaryocytic lineage are originated from hematopoietic stem/progenitor cells (HSPCs) and undergo differentiation, ultimately leading to the formation of

[1]Haihe Laboratory of Cell Ecosystem, State Key Laboratory of Experimental Hematology, National Clinical Research Center for Geriatric Disorders, Key Laboratory for Carcinogenesis and Invasion, Chinese Ministry of Education, Key Laboratory of Carcinogenesis, Chinese Ministry of Health, Department of Hematology, Xiangya Hospital, Central South University, Changsha, Hunan, China. [2]Cancer Research Institute, School of Basic Medical Science, Central South University, Changsha, Hunan, China. [3]State Key Laboratory of Experimental Hematology, National Clinical Research Center for Blood Diseases, Haihe Laboratory of Cell Ecosystem, Institute of Hematology & Blood Diseases Hospital, Chinese Academy of Medical Sciences & Peking Union Medical College, Tianjin, China. [4]Department of Orthopaedic Surgery, Second Xiangya Hospital, Central South University, Changsha, Hunan, China. [5]Xiangya School of Public Health, Central South University, Changsha, Hunan, China. [6]Department of Hematology, Peking Union Medical College Hospital, Chinese Academy of Medical Sciences and Peking Union Medical College, Beijing, China. [7]These authors contributed equally: Chunmei Kuang, Meijuan Xia.
✉ e-mail: zhoujx@ihcams.ac.cn; wenzhou@csu.edu.cn

platelets[7]. It has been reported that thrombocytopenia is caused by diminished HSPCs in the BM, defective megakaryocyte differentiation, unfavorable BMME, and/or immune-mediated platelet apoptosis[8–11]. Recently, we demonstrated that glycine concentration in the BMME is elevated and contributes to MM progression by disrupting glutathione (GSH) balance[12]. Meanwhile, we also found the intestinal bacteria promote MM progression by elevating the level of BM glutamine[13] and phosphoglycerate dehydrogenase (PHGDH), the first rate-limiting enzyme of the serine synthesis pathway (SSP), and that glutamine enhances cell proliferation and BTZ resistance of MM by increasing GSH synthesis in MM cells[14]. Thus, the alterations of metabolite levels in the BMME can accelerate MM. It has also been shown that metabolites such as cholesterol and serotonin are involved in megakaryopoiesis and thrombopoiesis during HSPC differentiation, suggesting that the uptake of metabolites might play a role in MK differentiation[15,16]. However, whether MK differentiation and platelet production from HSPCs are impaired in MM and what metabolites are involved in these processes remain to be defined.

In this work, we determine the clinical impact of thrombocytopenia in patients with MM at diagnosis. We also examine the differential metabolites in the BMME of MM patients with thrombocytopenia and the effects of metabolites on megakaryopoiesis and thrombopoiesis. We further investigate the role and molecular mechanism by which thrombocytopenia is induced by metabolites. Finally, we explore the therapeutic potential of intervening metabolite uptake in preventing thrombocytopenia.

## Results

### Thrombocytopenia is linked to poor prognosis in MM patients

To investigate the platelet counts and their potential correlation with disease progression in newly diagnosed MM (NDMM) patients, we studied 1468 patients and classified them into three groups based on the platelet counts in the peripheral blood (PB) at the time of diagnosis. The criterion of thrombocytopenia was defined as an absolute platelet count ≤$100 \times 10^9$/L or ≤$150 \times 10^9$/L in PB according to previous studies[3,17], while elevated platelet counts were defined as those higher than the upper limit of the normal count. Three groups were stratified as follows: normal platelet counts ($120–300 \times 10^9$/L), high platelet (>$300 \times 10^9$/L) and low platelet counts (≤$120 \times 10^9$/L). Among those patients, 906 (61.7%) had normal platelet counts, while 446 (30.4%) and 116 patients (7.9%) had low and high platelet counts, respectively (Fig. 1a).

We next compared the clinicopathologic parameters among the three groups, including gender, age, ISS (International Staging System) stage, DS (Durie -Salmon) stage, percentages of plasma cells, renal dysfunction, serum lactate dehydrogenase (LDH), serum calcium and IgH translocations, to explore whether platelet counts correlated with disease progression and may serve as a prognostic marker in MM patients. We found that platelet counts had no correlation with age, renal dysfunction, and serum calcium (Table 1). Intriguingly, we observed strong correlation between the platelet numbers and ISS stage, as evidenced by the significant difference in platelet counts in patients among three different stages. Specifically, the patients in stage I had the highest number of platelets, while the patients in stage III showed the lowest platelet count (Fig. 1b). Moreover, this correlation may be explained by the increased proportion of MM patients with thrombocytopenia at stage III of ISS (Supplementary Fig. 1a). In addition, we found that the platelet counts were also associated with sex, DS stage, the percentage of plasma cells in BM, the level of hemoglobin (Hb), the level of LDH, 1q21 gain, TP53 deletion and IgH translocation (Table 1).

To further assess whether platelet counts were associated with overall survival (OS) and progression-free survival (PFS), we conducted Kaplan-Meier analysis of the MM patients. Although the risk stratification of MM is widely available in the clinics[17], we found that both OS

and PFS were significantly worsened once the platelet counts were lower than $120 \times 10^9$/L. In contrast, no difference for OS and PFS was observed between patients with platelet count higher than $300 \times 10^9$/L and those within $120–300 \times 10^9$/L (Supplementary Fig. 1b). We next performed multivariable Cox regression analysis to investigate the prognostic value of platelet counts in MM patients. We found that low platelet counts, 1q21 gain, and TP53 deletion are independent prognostic factors for MM patients (Supplementary Table 1). We further combined the analysis of platelet count with the ISS stages. Not surprisingly, platelet counts showed minimal correlation with OS in patients with ISS stage I. However, OS were significantly shorter when the platelet count was lower than $120 \times 10^9$/L in patients of stage II and III (Supplementary Fig. 1c). To analyze the patients with lower platelet counts, we divided them into two groups with platelet counts $50–120 \times 10^9$/L and <$50 \times 10^9$/L, respectively. Strikingly, we found that patients with <$50 \times 10^9$/L platelet counts exhibited extremely inferior OS and PFS when compared with those with normal platelet counts. The most significant difference was observed in MM patients with ISS stage III (Fig. 1c, d). Thus, consistent with previous observations, the lower platelet count (or thrombocytopenia) is tightly linked to the progression and the outcome of MM. We also demonstrated that the lowest platelet count (<$50 \times 10^9$/L) predicts the poorest OS and PFS. Thus, low platelet counts may be used as a marker for unfavorable outcomes for MM patients.

It is well known that MKs are originated from HSPCs and MKs extend long branching processes into sinusoidal blood vessels to release platelets within the BM. It is therefore possible that MKs may be also impaired during MM progression and cause the reduction in platelet counts. We thus examined the number of MKs in MM patients of different stages or with various platelet counts with staining of Wright-Giemsa, leading us to discover a gradual decrease of MK numbers in MM patients compared with the healthy donors (Supplementary Table 2). Specifically, the patients at stages II and III showed significantly lower amounts of MKs (Supplementary Fig. 1d). Furthermore, fewer MKs were found in the patients with lower count of platelets, and the number of MKs and platelets correlated positively (r = 0.3266, p = 0.0004) (Supplementary Fig. 1e, f). In addition, we further determined the number of CD41+ cells in bone tissues of MM patients with immunostaining (Fig. 1e). Consistent with the results we observed in bone aspirates, there were lower MKs in bone tissues of MM patients with platelets less than $120 \times 10^9$/L (Fig. 1f), while the number of MKs and platelets also correlated positively (Fig. 1g, r = 0.5141, p = 0.0004). Thus, megakaryopoiesis is also impaired in MM patients.

We further examined the dynamic changes of platelet counts during MM progression in vivo by taking advantage of a previously described MM mouse model[13,18]. We measured complete blood counts every week after the injection of luciferase-expressing 5TGM1 mouse myeloma cells. MM progression in the mice was monitored weekly via the determination of tumor burden, including bioluminescent imaging, and the concentration of serum IgG2b (Fig. 1h). As expected, increased tumor infiltration and gradual elevation of serum IgG2b level were observed during the progression of MM (Supplementary Fig. 1g and Fig. 1i). Interestingly, the gradual decrease of platelets was seen in parallel with the elevation of serum IgG2b level (Fig. 1i). Moreover, platelet counts correlated strongly with IgG2b level (r = 0.52, p <0.0001) (Fig. 1j).

We next measured the distributions of MKs in the BM by conducting immunostaining of CD41, a widely used MK surface marker. Indeed, much fewer CD41+ MKs were observed in 5TGM1 mice than in the control mice (Fig. 1k). The number of MKs was also significantly lower in 5TGM1 mice with low platelet counts than those with normal platelet counts (Supplementary Fig. 1h, i). Meanwhile, the number of MKs in BM was reduced during disease progression, while a striking decrease was observed when the level of IgG2b was above 6 mg/mL

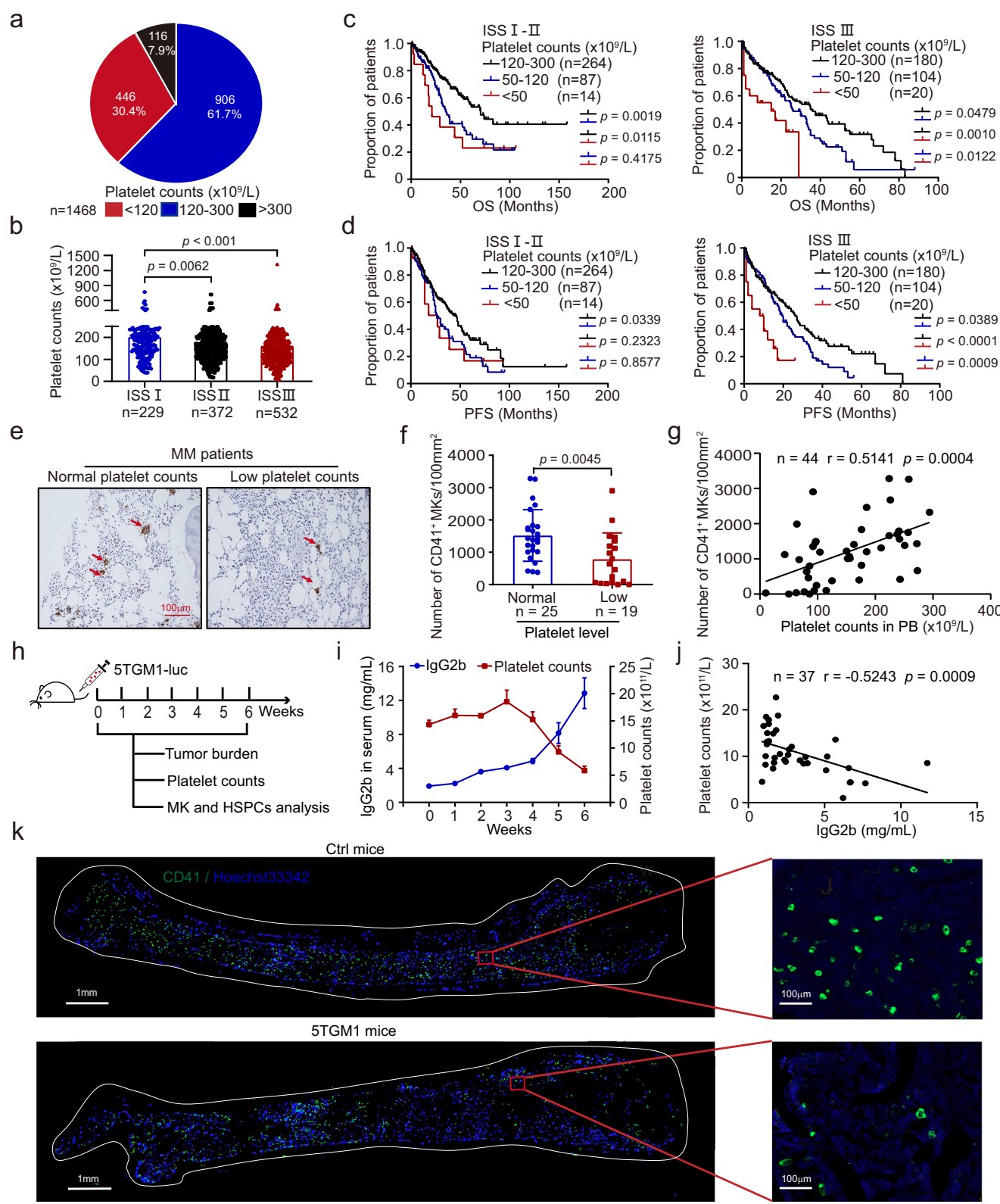

(Supplementary Fig. 1j, k). We also measured the number of the progenitor cells along the megakaryocytic lineages, including LSK, CMP, and MEP, and observed a decrease in these MK progenitors (Supplementary Fig. 1l), especially in mice with IgG2b above 6 mg/mL (Supplementary Fig. 1m). Together, analyses of MM patients and experiments with the 5TGM1 mice model suggest that megakaryopoiesis and thrombopoiesisis severely impaired and leads to thrombocytopenia during MM progression.

## Serine is upregulated in the BM and linked to thrombocytopenia in MM

The production of platelets by BM MKs which are originated from HSPCs tightly depends on their interactions with the BM niche. We, therefore, asked whether the defects in megakaryopoiesis and thrombopoiesis that occur in the BM of MM patients were caused by soluble factors in the niche. To test this, we took advantage of a previously established in vitro culture system to model human

**Fig. 1 | Thrombocytopenia is linked to poor prognosis in MM patients.**
**a** Assessment of platelet counts in newly diagnosed MM (NDMM) patients (n = 1,468). Three groups were defined as follows: patients with normal-platelet counts (120–300 × 10⁹/L, n = 906), high-platelet counts (> 300 × 10⁹/L, n = 116), and low-platelet counts (≤120 × 10⁹/L, n = 446). **b** Platelet counts in PB (peripheral blood) of NDMM patients with different ISS stage (mean ± SD, nMM (ISS I) = 229, nMM (ISS II) = 372; nMM (ISS III) = 532). **c, d** Kaplan-Meier analyses of overall survival (OS) and progression free survival (PFS) in NDMM patients with normal-platelet counts (120–300 × 10⁹/L, nMM (ISS I- II) = 264, nMM (ISS III) = 180), low-platelet counts (50–120 × 10⁹/L, nMM (ISS I- II) = 87, nMM (ISS III) = 104) and low-platelet counts (<50 × 10⁹/L, nMM (ISS I- II) = 14, nMM (ISS III) = 20). **e** Representative images of CD41⁺ MKs in BM biopsies of MM patients with normal platelet counts or low platelet counts by using immunohistochemical staining (scale bar, 100μm). **f** Statistical analysis of the number of CD41⁺ MKs per 100 mm² in the BM biopsies of MM patients with normal platelet counts (n = 25) or low platelet counts (n = 19).

Results represent means ± SD. **g** Assessment of the correlation between platelet counts and numbers of CD41⁺ MKs in BM (n = 44). **h** Schematic diagram of the experimental design to detect platelet counts, tumor burden, and the proportion of LSKs, CMPs, and MEP in 5TMG1 mice. **i** Dynamic changes of IgG2b levels during MM progression in 5TGM1 mice as detected with ELISA. Peripheral platelet counts of 5TGM1 mice were measured with haematology analyser (n = 6). **j** Assessment of the correlation between IgG2b levels and platelet counts in 5TGM1 mice (n = 37). **k** Representative images of immunofluorescence analysis for MKs staining using CD41-FITC antibody (Green) and Hoechst 33342 (Blue) in the BM of control mice (Ctrl mice) and 5TGM1 mice. Top, normal numbers of MKs in ctrl mice; bottom, decreased numbers of MKs in 5TGM1 mice. The scale bars are 1 mm and 100 μm for the left image and right image, respectively. Results represent means ± SD. An unpaired two-sided Student's t-test was used to assess the difference in **b, f**; Two-sided log-rank (Mantel-Cox) test were used in **c, d**; Two-sided Pearson test were used in **g, j**. Source data are provided as a Source Data file.

## Table 1 | The correlation of platelet count and clinical characteristics of MM

| Patients' Characteristics | Low platelet count (n = 446) (n/N × 100%) | Normal platelet count (n = 906) (n/N × 100%) | High platelet count (n = 116) (n/N×100%) | p-value (low vs. normal) |
|---|---|---|---|---|
| Age (y) | 60 | 59 | 57 | 0.056[a] |
| Sex | | | | 0.049[b]* |
| Male | 292/446 (65.5) | 543/906 (59.9) | 62/116 (53.4) | |
| Female | 154/446 (34.5) | 363/906 (40.1) | 54/116 (46.6) | |
| ISS stage | | | | 0.000[b]*** |
| I | 42/334 (12.6) | 165/715 (23.1) | 22/84 (26.2) | |
| II | 96/334 (28.7) | 248/715 (34.7) | 28/84 (33.3) | |
| III | 196/334 (58.7) | 302/715 (42.2) | 34/84 (40.5) | |
| DS stage | | | | 0.001[b]** |
| I | 13/400 (3.3) | 59/768 (7.7) | 8/97(8.2) | |
| II | 30/400 (7.5) | 86/768 (11.2) | 15/97(15.5) | |
| III | 357/400 (89.2) | 623/768 (81.1) | 74/97(76.3) | |
| Plasma cells in BM (%) | 39.7 | 31.98 | 24.3 | 0.000[a]*** |
| Median Calcium (mmol/L) | 2.31 | 2.34 | 2.31 | 0.056[a] |
| Renal dysfunction | 98/310 (31.6) | 165/785 (21) | 19/99 (19.2) | 0.236[b] |
| Median Hb (g/L) | 75 | 95 | 96 | 0.000[a]*** |
| Median LDH (U/L) | 221 | 168.1 | 161 | 0.000[a]*** |
| 1q21 gain | | | | 0.004[b]** |
| No | 64/172 (37.2) | 214/419 (51) | 30/46 (65.1) | |
| Yes | 108/172 (62.8) | 205/419 (49) | 16/46 (34.9) | |
| TP53 deletion | | | | 0.037[b]* |
| No | 136/175 (77.7) | 381/457 (83.4) | 50/54 (92.6) | |
| Yes | 39/175 (22.3) | 76/457 (16.6) | 4/54 (7.4) | |
| IgH translocation | | | | 0.046[b]* |
| No | 64/180 (35.6) | 193/436 (44.3) | 22/52 (42.3) | |
| Yes | 116/180 (64.4) | 243/436 (55.7) | 30/52 (57.7) | |

*DS stage* Durie-Salmon stage, *ISS* International Staging System, *FISH* fluorescence in situ hybridization, *LDH* lactate dehydrogenase, *y* years. *Hb* hemoglobin, Renal dysfunction: Creatinine level > 2.0 mg/dL; 1q21 gain: FISH 1q21 gain in ≥10% of cells; TP53 deletion: FISH TP53 deletion in ≥10% of cells; FISH IgH translocation in ≥10% of cells; *OS* overall survival, *PFS* progression-free survival; all MM patients were newly diagnosed. *p < 0.05, **p < 0.01, ***p < 0.001. [a]Unpaired two-sided Student's t-test. [b]Two-tailed Pearson Chi-Square test.

megakaryopoiesis and thrombopoiesis[19]. We assessed MK differentiation and platelet production of HSPCs (CD34⁺ cells) incubated with either the BM plasma associated with normal platelet counts or with low platelet counts (Fig. 2a). Interestingly, we found that the fraction of CD41a⁺CD42b⁺ cells was significantly reduced with the treatment of BM plasma of the low platelet counts (p = 0.0498) (Fig. 2b). The formation of proplatelets and CD41a⁺CD42b⁺ platelet-like particles (PLPs) was also significantly impaired with the treatment of BM plasma of low platelet counts (p = 0.0027) (Fig. 2c, d).

Recently, we have validated that gut microbiome and PHGDH promotes MM progression via de novo synthesis of glutamine and serine, respectively[13]. To identify the potential metabolic factor(s) that mediate the inhibitory effects of myeloma cells, we first conducted untargeted metabolomics analysis of the BM plasma collected from the MM patients using chromatography-mass spectrometry. We detected approximately 200 metabolites that cover a wide range of the metabolic pathways. Among those, 16 metabolites, including both amino acids and lipids, exhibited significant differences between patients with normal platelet counts and low platelet counts (Fig. 2e). Interestingly, 5 amino acids (glycine, L-serine, L-aspartic acid, L-glutamic acid, and ornithine) were found to accumulate more in patients with lower platelet counts (Fig. 2f). To gain further insights into the changes of metabolites, we determined the levels of amino acids by using Liquid Chromotography Mass Spectrometry and found that the levels of glycine, L-serine, and L-aspartic acid were much higher in the BM of patients with low platelet counts (Fig. 2g, h). We also measured the level of the 3 amino acids in the plasma of 5TGM1 and control mice, allowing us to discover that the levels of glycine and L-serine were higher in 5TGM1 mice. In contrast, L-aspartic acid exhibited no significant difference between the two groups (Fig. 2i).

We next asked whether serine and glycine impacted megakaryopoiesis and thrombopoiesis in vitro. We treated CD34⁺ cells with these amino acids individually using the MK differentiation system as described earlier in this study. While glycine exerted minimal effects on megakaryopoiesis and thrombopoiesis, serine potently suppressed megakaryopoiesis and thrombopoiesis (Fig. 2j–l). Together, experiments with metabolic profiling and functional validation allowed us to reveal serine as a potential key factor in the BM microenvironment to control megakaryopoiesis and thrombopoiesis.

## Serine inhibits megakaryopoiesis and thrombopoiesis both in vitro and in vivo

We next determined whether serine functioned to suppress megakaryopoiesis and thrombopoiesis. First, we added increasing concentrations of serine to the MK differentiation system in vitro, which led to elevating levels of intracellular serine in the cells, suggesting that the uptake of extracellular serine of MKs was dose-dependent

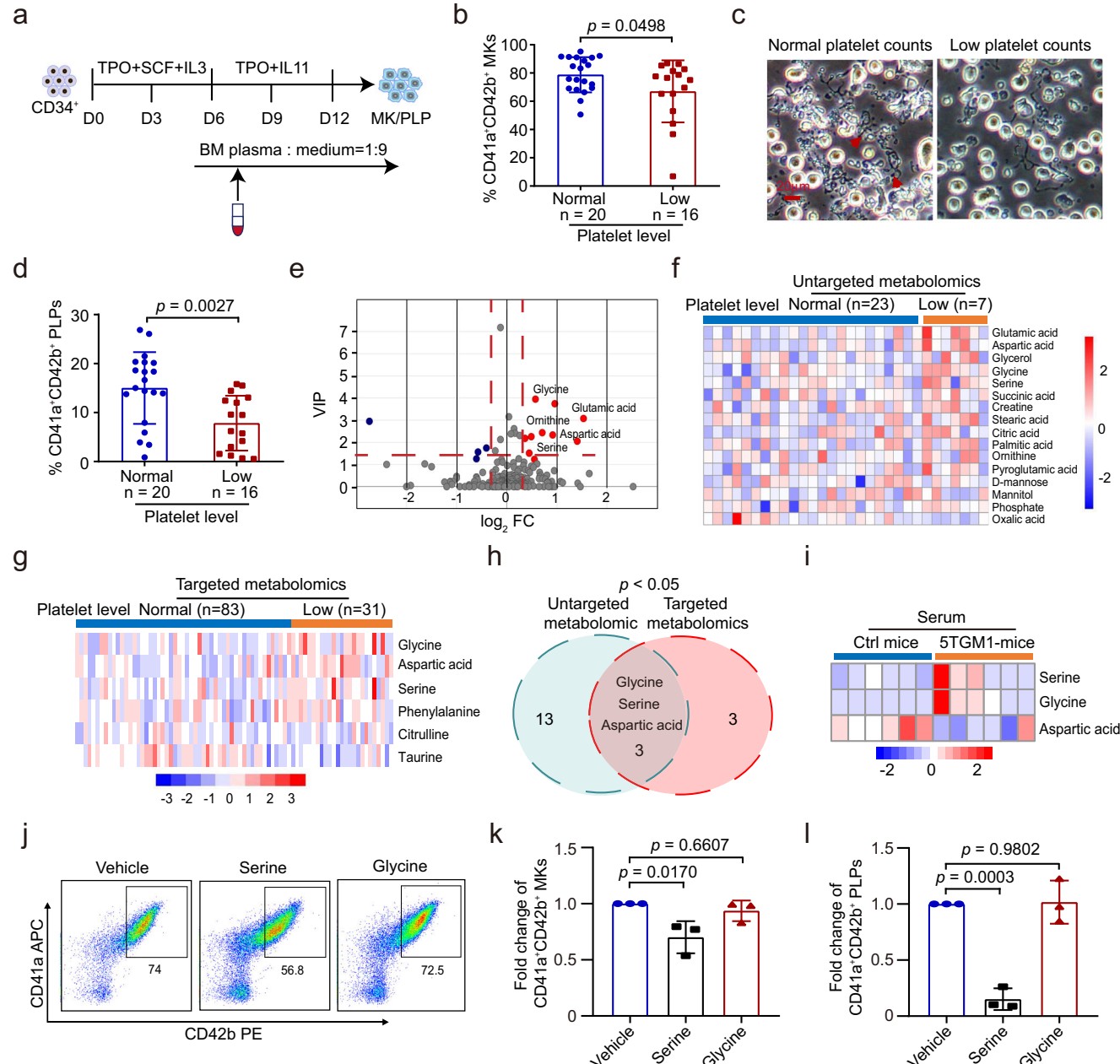

**Fig. 2 | Serine is upregulated in BM and linked to thrombocytopenia in MM.**
**a** Schematic diagram showing megakaryocytic differentiation of CD34[+] cells trea-ted with BM plasma derived from MM patients with or without thrombocytopenia.
**b** Flow cytometry analysis of the percentage of CD41a[+]CD42b[+] cells at day 12 of megakaryocytic differentiation with BM plasma treatment (mean ± SD, $n$MM$_{normal}$ = 20, $n$MM$_{low}$ = 16). **c** Phase-contrast images of proplatelet formation at day 12 from CD34[+] cells with BM plasma treatment (scale bar, 20μm). **d** Generation of CD41a
[+]CD42b[+] PLPs at day 12 of CD34[+] cells with BM plasma treatment (mean ± SD).
**e** Volcano plot of differential metabolites in BM plasma between MM patients with normal-platelet counts ($n$ = 23) and patients with low-platelet counts ($n$ = 7) by using with un-targeted metabolomics. Significantly upregulated metabolites are represented as 'red' dots, and downregulated metabolites are represented as 'blue' dots. **f** Heatmaps of the 16 differential metabolites in BM plasma between MM patients with normal platelet counts and with low platelet counts. **g** Heatmaps of

the 6 differential amino acids in BM plasma between MM patients with normal platelet counts ($n$ = 83) and patients with low platelet counts ($n$ = 31) revealed with targeted metabolomics. **h** Venn diagram of un-targeted metabolomics and targeted metabolomics. **i** Heatmap of serine, glycine, and aspartic acid in serum between control (ctrl) mice and 5TGM1 mice ($n$ = 6). **j** Representative plots for flow cyto-metry detection of the proportion of CD41a[+]CD42b[+] cells differentiated from CD34[+] cells treated with vehicle, serine, and glycine. **k** Statistical analysis of the fold change of CD41a[+]CD42b[+] cells in differentiated cells treated with vehicle, serine and glycine (mean ± SD, $n$ = 3 independent experiments). **l** Statistical analysis of the fold change of CD41a[+]CD42b[+] PLPs in differentiated cells treated with vehicle, serine, and glycine (mean ± SD, $n$ = 3 independent experiments). Results represent means ± SD. Unpaired two-sided $t$-test were used in **b**, **d**. One-way ANOVA followed by Dunnett's multiple comparison test was used in **k**, **l**. Source data are provided as a Source Data file.

(Supplementary Fig. 2a). We then discovered that at the serine dose of 2 mM, MK differentiation and platelet generation were inhibited, while minimal effects were seen when serine was under 0.5 mM. At 8 mM of serine, the inhibition was highly potent (Fig. 3a–c). However, the ploidy

of MKs is not affected by serine, suggesting that serine does not affect the maturation of MKs (Supplementary Fig. 2b). Surprisingly, the apoptotic rate of MKs was enhanced by excessive serine (Supple-mentary Fig. 2c).

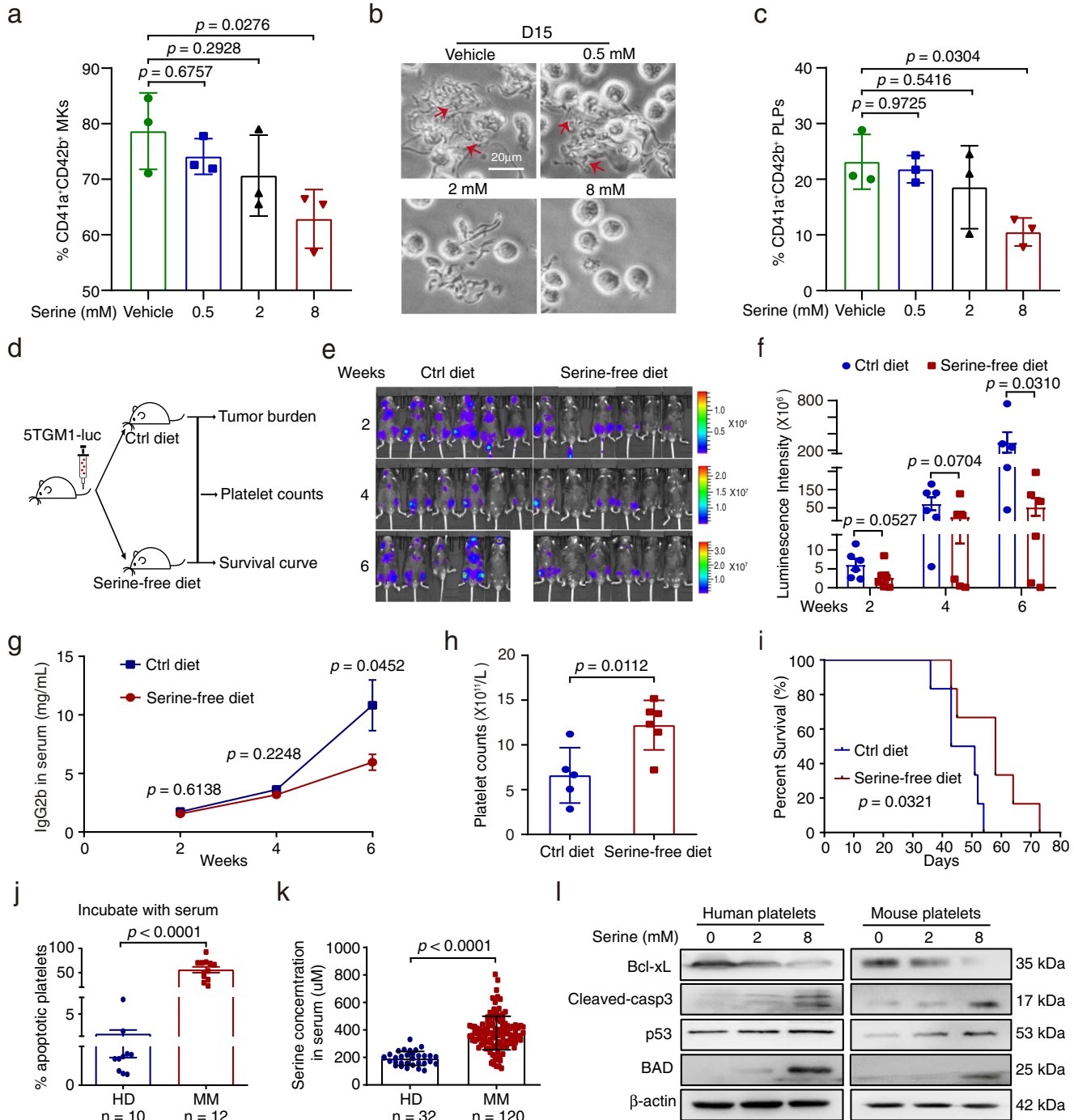

**Fig. 3 | Serine inhibits megakaryopoiesis and thrombopoiesis both in vitro and in vivo. a** The production of CD41a⁺CD42b⁺ MKs of CD34⁺ cells treated with indicated concentration of serine (mean ± SD, *n* = 3 independent experiments). **b, c** Phase-contrast images of proplatelet formation and the production of CD41a⁺CD42b⁺ PLPs at day 15 from CD34⁺ cells treated with indicated concentrations of serine (mean ± SD, *n* = 3 independent experiments). The scale bar is 20 μm. **d** Schematic diagram of the experimental design to detect tumor burden, platelet counts and the survival in 5TMG1 mice fed with the control diet (ctrl diet) or serine-free diet (*n* = 6 per group). **e, f** Tumor-associated luminescence intensity and quantification of luminescence intensity in 5TGM1 mice fed with the ctrl diet or serine-free diet at 2, 4, and 6 weeks (Week 2 and 4: *n* = 6 in each group, Week 6: *n* = 5 in ctrl diet group, *n* = 6 in serine-free diet group). Results represent means ± SEM. An unpaired one-sided Student's *t*-test was used in **f. g** The concentrations of IgG2b in mouse serum as detected with ELISA

(Week 2 and 4: *n* = 6 in each group, Week 6: *n* = 5 in ctrl diet group, *n* = 6 in serine-free diet group). **h** The number of platelet counts in PB in 5TGM1 mice at week 6 (*n* = 5 in ctrl diet group, *n* = 6 in serine-free diet group, mean ± SD). **i** The survival curves of 5TGM1 mice fed with the ctrl diet or serine-free diet. A two-sided log-rank (Mantel-Cox) test was used to assess the difference. **j** The apoptosis rate of platelets derived from HD incubated with serum from HD or MM patients (nHD = 10, nMM = 12, mean ± SEM). **k** The level of serine in serum from HD or MM patients as revealed with targeted metabolomics (nHD = 32, nMM = 120, mean ± SD). **l** Western blot analysis of Bcl-xL, C-caspase3, p53, BAD and β-actin expression in platelets derived from HD or control mice treated with indicated concentration of serine. Experiment was repeated independently for 3 times. An unpaired two-sided Student's *t*-test was used in **g, h, j, k**; One-way ANOVA followed by Dunnett's multiple comparison test in **a, c**. Source data are provided as a Source Data file.

We also assessed the kinetic pattern of serine levels (in week 0, 3 and 6) in the PB serum of 5TGM1 mice in vivo. Consistent with the reduced platelet counts in 5TGM1 mice in week 6, the level of serine was higher in PB serum of the 5TGM1 mice than the control mice, while little difference was observed in 5TGM1 mice and control mice in week 3 (Supplementary Fig. 2d). Furthermore, the concentration of serine in the serum correlated inversely with platelet counts (Supplementary Fig. 2e).

After assessing the function of serine using the in vitro assay, we further elucidated its effects in vivo. We developed MM in mice by injecting 5TGM1 cells and then fed them with diets with or without serine (Fig. 3d). The two diets caused significant differences in the level of serine in vivo. After the depletion of serine from the diet, the level of serine in PB serum clearly decreased (Supplementary Fig. 2f). We further measured the dynamic changes of tumor cells in vivo to monitor the tumor burden. We found that the number of infiltrated tumor cells was smaller in mice with serine-free diet than the control mice ($p <0.05$) (Fig. 3e, f). We then compared the level of IgG2b in serum of 5TGM1 mice with or without the serine diet. While minor difference in IgG2b level was observed after 3 weeks of diet, the difference became quite significant after 5 weeks of uptake (Fig. 3g). Consistent with the results from the in vitro experiments, the platelet count and MKs in the mice fed with serine-free diet were sustained at a higher level, and the deprivation of serine uptake indeed prevented the decrease of platelets during MM progression ($p <0.05$) (Fig. 3h). However, the proportion of LSK, CMP, and MEP showed minimal differences between these two groups (Supplementary Fig. 2g). Finally, we found that the deprivation of serine from the diet significantly extended the survival time ($p = 0.0321$) (Fig. 3i). Thus, the limitation of serine uptake suffices to delay MM progression in vivo.

It was previously reported that an increased apoptotic rate of platelets is another cause of thrombocytopenia[20,21]. These findings prompted us to determine whether the rate of platelet apoptosis was elevated in MM patients. We incubated platelets with the PB serum from either healthy donors or MM and found that the rate of apoptosis was significantly higher in platelets incubated with the serum from MM ($p < 0.0001$) (Fig. 3j). Moreover, the rate of apoptosis was significantly higher in platelets incubated with the serum with a higher level of serine from MM patients (Supplementary Fig. 2h). These findings suggest the existence of pro-apoptotic factors in the serum of MM patients. In separate experiments, we also confirmed the increase of serine levels in the serum of MM patients using targeted metabolic analysis ($p < 0.0001$) (Fig. 3k). Furthermore, the addition of serine enhanced platelet apoptosis, as indicated by the flow cytometry analysis (Supplementary Fig. 2i, j).

At the molecular level, we found that serine increased caspase3 cleavage and enhanced the expression of p53 and BAD, while reducing the Bcl-xL level in platelets isolated from healthy donors and control mice (Fig. 3l). Moreover, MM patients with high levels of serine and thrombocytopenia were associated with ISS stage and the levels of Hb (Supplementary Table 3). Thus, high levels of serine induce apoptosis of platelets in MM.

## Serine uptake is mediated by SLC38A1

To explore the potential downstream target genes of serine, we performed RNA-sequencing analysis of CD34+ cells undergoing megakaryocytic differentiation treated with vehicle or serine and found that a total of 3,247 genes were upregulated while 2,002 were downregulated in cells treated with serine (Supplementary Data 1). Among those, *SLC38A1*, *MTHFD1*, *GCSH*, and *PHGDH* were upregulated (Fig. 4a), suggesting that exogenous serine might induces alterations in serine transport, folate cycle, conversion of serine to glycine, and serine synthesis (Fig. 4b). Furthermore, increase of *SLC38A1* mRNA and protein levels were observed in serine-treated cells (Fig. 4c, d). To explore whether the uptake of exogenous serine is mediated by the amino transporter, we depleted *SLC38A1* by using shRNAs in CD34+ cells via lentiviral infection (Fig. 4e, f), leading to the increase of serine levels in the conditioned medium (CM) from cells with *SLC38A1* knockdown (Fig. 4g). Furthermore, knockdown of *SLC38A1* enhanced megakaryocytic differentiation (Fig. 4h) and PLP generation (Fig. 4i, j). Thus, SLC38A1 serves as a critical transporter mediating the inhibitory effects of serine on megakaryopoiesis and thrombopoiesis. Restriction of serine uptake may therefore serve as a potential effective strategy for improving thrombocytopenia.

## Serine downregulates Supervillin via S-adenosyl-methionine-mediated tri-methylation of H3K9

To further dissect the molecular mechanism by which serine regulates megakaryopoiesis and thrombopoiesis, we determined the fate of the carbons in serine. We cultured CD34+ cells undergoing MK differentiation in media containing $^{13}C_3$-serine (Supplementary Fig. 3a) and found that serine-derived carbons were incorporated into glycine, methionine, S-adenosyl-methionine (SAM), S-adenosylhomocysteine (SAH), and nucleotides in cells undergoing MK differentiation (Supplementary Fig. 3 and Supplementary Data 2). Because labeled methionine, SAM, and SAH were detected stably in our system (Fig. 5a), we speculated that serine is likely incorporated into the methionine cycle through the folate one-carbon metabolism (Fig. 5b). We next examined the effects of methionine on megakaryocytic differentiation and PLP generation, leading us to find that methionine also decreased megakaryocytic differentiation and PLP generation (Fig. 5c, d). To assess whether the effect of serine was methionine-dependent, we knocked down serine transporter *SLC38A1* in CD34+ cells and examined the effects in the presence of vehicle or serine. We found that knock-down of *SLC38A1* promotes megakaryocytic differentiation and PLP generation. However, the addition of methionine prevented the effect of *SLC38A1* knock-down, which indicates that the effects of serine is methionine-dependent (Supplementary Fig. 3c, d). To further explore whether excessive serine inhibits megakaryocytic differentiation and PLP generation through one-carbon unit metabolism, we knocked down *SHMT2*, which is required for the conversion of serine into glycine and a tetrahydrofolate-bound one-carbon unit. Interestingly, the knock-down of *SHMT2* promotes megakaryocytic differentiation and PLP generation (Fig. 5e, f).

Next, we analyzed the contribution of serine-dependent one-carbon metabolism to the methionine cycle and the function of SAM in differentiated cells. We first infected CD34+ cells with small hairpin RNAs (shRNAs) that targeted *MAT2A* (Supplementary Fig. 3e, f), which inhibits the conversion of methionine to SAM. We found that the knock-down of *MAT2A* promotes megakaryocytic differentiation and PLP generation (Supplementary Fig. 3g, h). We further added 3-Deazaa-denosine (3DZA), an inhibitor that suppresses S-adenosylhomocysteine hydrolase and SAM-dependent methylation reactions, to CD34+ cells undergoing megakaryocytic differentiation. We found that the inhibition of SAM-dependent methylation reactions promotes megakaryocytic differentiation and PLP generation (Fig. 5g, h), which further indicates that the effect is SAM-dependent. We assessed the expression of methyltransferases and found that *SETDB2* and *NSD1* were significantly upregulated upon serine stimulation (Supplementary Fig. 4a). We then examined the levels of H3K9me3 and H3K36me3 after serine stimulation. As expected, the levels of H3K9me3 and H3K36me3 were elevated in cells treated with serine (Fig. 5i). Because SETDB2 is an H3K9 methyltransferase that catalyzes H3K9me3 to repress gene expression[22], we focused on the effects of serine on H3K9me3 expression in later study. The level of H3K9me3 was elevated in cells treated with methionine (Supplementary Fig. 4b). Meanwhile, *SHMT2* knock-down significantly inhibits the expression of H3K9me3, suggesting that the elevation of H3K9me3 is mediated by the one-carbon metabolism (Supplementary Fig. 4c).

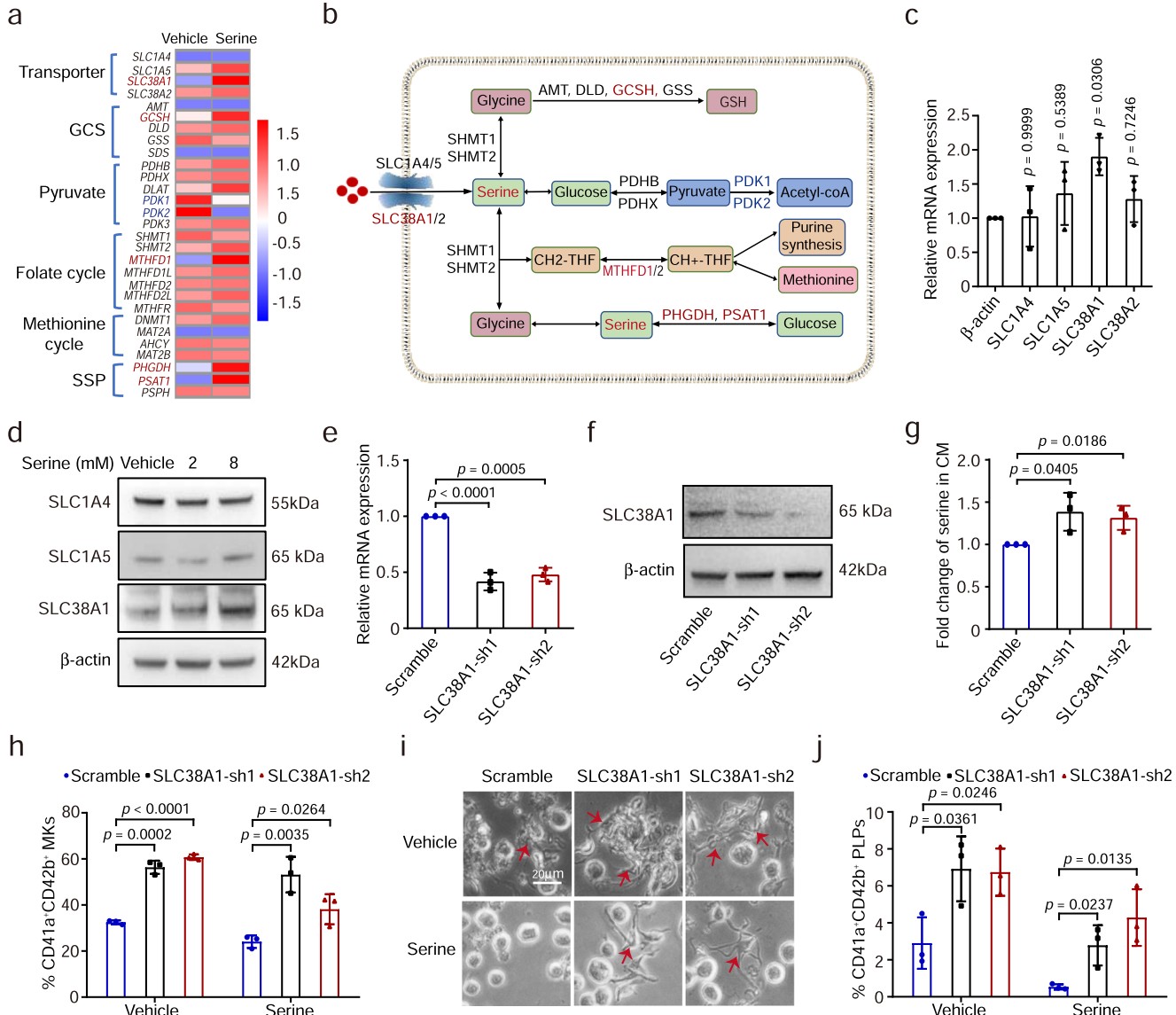

**Fig. 4 | Serine uptake is mediated by SLC38A1. a** The expression profile of serine metabolism-related genes in MKs cultured with vehicle or excessive serine. **b** Schematics of serine metabolism. **c** Assessment of mRNA expression of serine transporters in differentiated cells treated with vehicle or excessive serine at day 12 (mean ± SD, $n = 3$ independent experiments). Significance was analyzed with an One-way ANOVA followed by Dunnett's multiple comparison test. **d** Protein levels of SLC1A4, SLC1A5 and SLC38A1 were detected with immunoblotting in differentiated cells treated with vehicle or serine at day 12. $n = 3$ independent experiments. **e** The mRNA levels of *SLC38A1* in CD34[+] cells containing control and *SLC38A1* shRNAs as detected with qPCR analysis (mean ± SD, $n = 3$ independent experiments). **f** Protein levels of SLC38A1 were detected with immunoblotting in CD34[+] cells infected with lentivirus-containing scramble or SLC38A1-shRNA. $n = 3$

independent experiments. **g** The concentration of extracellular serine in CD34[+] cells containing control and *SLC38A1* shRNAs (mean ± SD, $n = 3$ independent experiments). **h** Flow cytometry analysis of the percentage of CD41a[+]CD42b[+] MKs from CD34[+] cells with SLC38A1 knockdown exposed to serine (mean ± SD, $n = 3$ independent experiments). **i** Representative images of proplatelet formation from MKs from CD34[+] cells with *SLC38A1* knockdown exposed to vehicle or serine. The scale bar is 20 μm. **j** Generation of CD41a[+]CD42b[+] PLPs at day 12 from CD34[+] cells with *SLC38A1* knockdown exposed to vehicle or serine (mean ± SD, $n = 3$ independent experiments). Results represent means ± SD. An unpaired two-sided Student's *t*-test was used in **e**, **g**, **h**, **j**. Source data are provided as a Source Data file.

To identify the potential methylated substrates regulated by serine, we performed assay for transposase-accessible chromatin using sequencing (ATAC-seq) in cells treated with vehicle or serine, respectively (Supplementary Data 3), allowing us to discover distinct pattern and numbers of peaks between control and serine-treated cells (Supplementary Fig. 4d). By integrating the downregulated peaks in ATAC-seq and the downregulated genes in RNA-seq, we identified 15 potential target genes potentially regulated by serine-induced H3K9me3 (Figure 5j), seven of which were further confirmed (Supplementary Fig. 4e), including Supervillin (*SVIL*), *SHANK3* and *FRYL* (Fig. 5k). To assess whether serine inhibits these genes through DNA methylation,

we first examined the methylation status of the promoter by using Methylation-Specific PCR (MS-PCR). However, there was no significant difference in DNA methylation upon serine stimulation (Supplementary Fig. 4f). Because H3K9me3 is a repressive mark that has been implicated in heterochromatin formation, we examined the chromatin statues in these three genes. Interestingly, differentially accessible peaks were identified in the promoters of *SVIL*, *FRYL*, and *SHANK3* (Fig. 5k). Next, we assessed the binding of H3K9me3 to the promoter of *SVIL*, *SHANK3*, and *FRYL* and observed enhanced binding of H3K9me3 to the promoter of *SVIL* and *SHANK3* upon serine stimulation, while the binding to *SVIL* promoter is stronger than *SHANK3* (Fig. 5l). At the

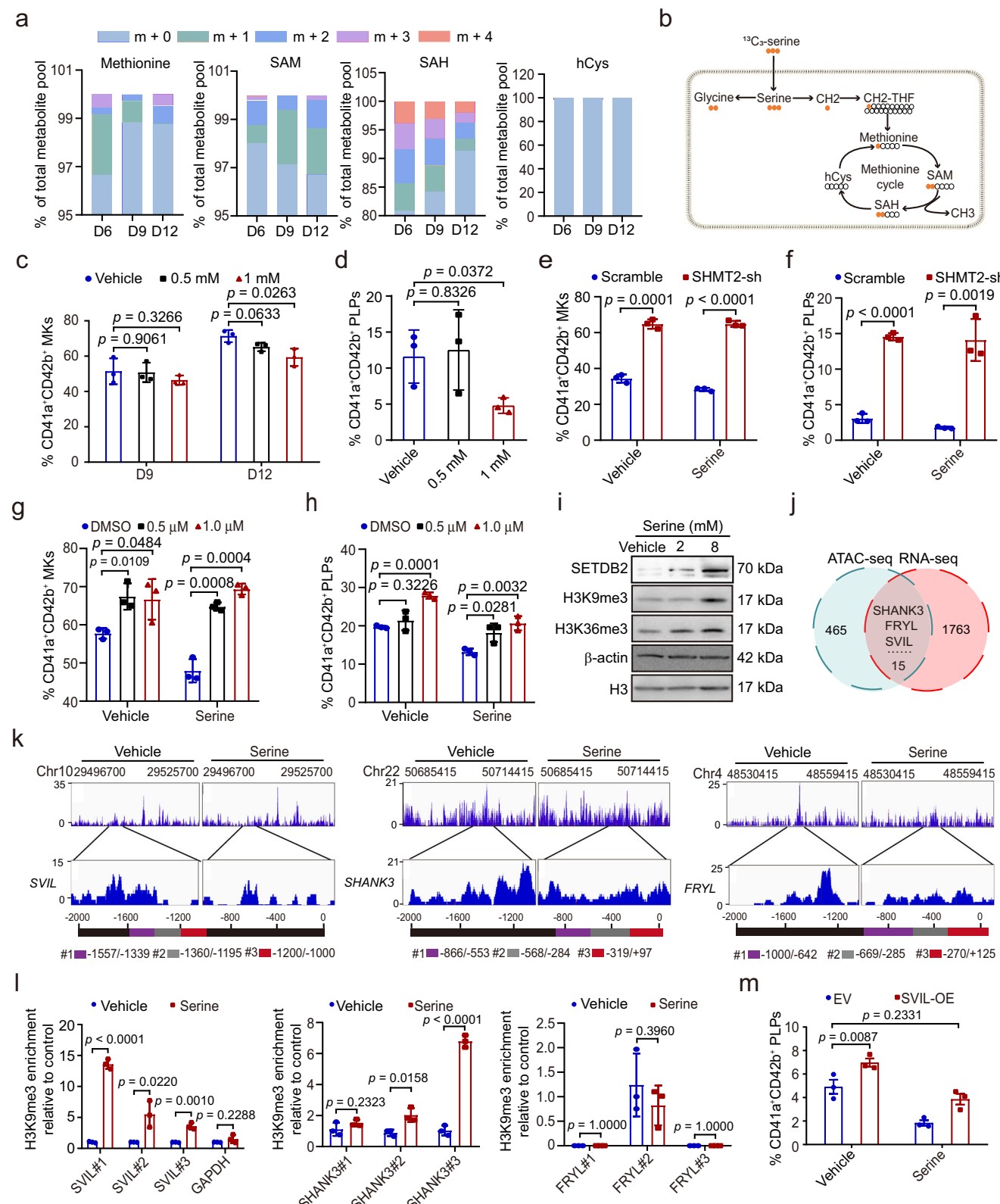

functional level, depletion of SVIL and SHANK3 inhibited mega-karyopoiesis and thrombopoiesis (Supplementary Fig. 4g–i), while overexpression of SVIL rescued the inhibitory effect of serine-mediated thrombocytopenia (Fig. 5m). To further assess whether the protein level of SVIL was also downregulated by serine, we performed Western blotting in differentiated cells treated with vehicle or serine. We found that SVIL protein was reduced in differentiated cells treated

with serine (Supplementary Fig. 4j). Consistence with the in vitro results, immunofluorescence analysis from 5TGM1 mice fed with serine-free diet also showed a lower level of SVIL than control group (Supplementary Fig. 4k). In contrast, *RUNX1*, a well-established regulator of megakaryocyte specification, maturation, and thrombopoiesis, was downregulated in the RNA-seq but not in the ATAC-seq analysis (Supplementary Fig. 4l, m). Because SETDB2 is responsible for

**Fig. 5 | Serine downregulates Supervillin via S-adenosyl-methionine-mediated tri-methylation of H3K9. a** The metabolic flux of $^{13}C_3$-serine in CD34$^+$ cells undergoing MK differentiation, m+0 denotes metabolites without labeled carbon atom, while m+1, m+2, m+3, and m+4 denotes metabolites with 1, 2, 3, and 4 labeled carbon atom. **b** Schematics of serine metabolism in MKs. **c** Generation of CD41a$^+$CD42b$^+$ MKs of CD34$^+$ cells with indicated concentrations of methionine (mean ± SD, $n = 3$ independent experiments). **d** Generation of CD41a$^+$CD42b$^+$ PLPs from CD34$^+$ cells with indicated concentrations of methionine (mean ± SD, $n = 3$ independent experiments). **e** Generation of CD41a$^+$CD42b$^+$ MKs from CD34$^+$ cells infected with scramble or SHMT2-shRNA virus with or without serine treatment (mean ± SD, $n = 3$ independent experiments). **f** Generation of CD41a$^+$CD42b$^+$ PLPs from CD34$^+$ cells infected with scramble or SHMT2-shRNA virus with or without serine treatment (mean ± SD, $n = 3$ independent experiments). **g** Generation of CD41a$^+$CD42b$^+$ MKs from CD34$^+$ cells treated with DMSO or the SAHH inhibitor 3DZA with or without serine treatment (mean ± SD, $n = 3$ independent

experiments). **h** Generation of CD41a$^+$CD42b$^+$ PLPs from CD34$^+$ cells treated with DMSO or the SAHH inhibitor 3DZA with or without serine treatment (mean ± SD, $n = 3$ independent experiments). **i** Western blot analysis of H3K9me3, H3K36me3, SETDB2 in cells at day 12 treated with vehicle or serine. Experiment was repeated independently for 3 times. **j** Venn diagram of downregulated genes and peaks in both ATAC-seq and RNA-seq analyses. **k** Peaks of the promoter of *SVIL*, *FRYL*, and *SHANK3* in differentiated cells cultured with vehicle or serine at day 12 as revealed with ATAC-seq. Different regions of primers used for ChIP-qPCR in the promoters are in different colors (purple, gray, and red). **l** The binding of H3K9me3 to the promoters of *SVIL*, *SHANK3*, and *FRYL* in 293T cells cultured with vehicle or serine by using ChIP-qPCR (means ± SD, $n = 3$ independent experiments). **m** Generation of CD41a$^+$CD42b$^+$ PLPs from CD34$^+$ cells with overexpression of SVIL exposed to serine (mean ± SD, $n = 3$ independent experiments). An unpaired two-sided Student's *t*-test was used in **c**, **d**, **e**, **f**, **g**, **h**, **l**, **m**. Source data are provided as a Source Data file.

catalyzation of H3K9me3, we then assessed the binding of H3K9me3 to the promoter of *SVIL* after knock-down of SETDB2 (Supplementary Fig. 4n). As expected, depletion of SETDB2 rescued the enhanced binding of H3K9me3 to the promoter of *SVIL* mediated by serine (Supplementary Fig. 4o). Together, these results suggest that serine inhibits megakaryopoiesis and thrombopoiesis via SVIL down-regulation, which is mediated by SAM-mediated tri-methylation of H3K9.

### Serine accumulated in the BM microenvironment is released from myeloma cell

We sought to investigate how serine is accumulated in the BM microenvironment. We previously demonstrated that PHGDH and PSPH catalyze the conversion of 3-phosphoglycerate to serine, leading us to assess *Phgdh* and *Psph* mRNA and protein levels in LK, B lymphocytes, plasma cells, T lymphocytes, erythroid cells, and BMSCs isolated from the BM of tumor-bearing 5TGM1 mice (Fig. 6a). Strikingly, we detected much higher levels of *Phgdh* mRNA and protein in the plasma cells than in other niche cells (Fig. 6b, c, Supplementary Fig. 5a). Moreover, the protein level of PHGDH is higher in CD138$^+$ cells derived from MM patients with low platelet count than those with normal platelet count (Supplementary Fig. 5b), suggesting that the PHGDH is upregulated in MM cells and may enhance the secretion of serine to the BM. In keeping with the notion, the fraction of plasma cells positively correlated with the serine level in the serum of 5TGM1 mice (Supplementary Fig. 5c). We, therefore, hypothesized that myeloma cells in the BM microenvironment might exert the inhibitory effects on megakaryopoiesis and thrombopoiesis by releasing excessive serine.

To test this hypothesis, we first determined the effects of applying the CM from MM cell culture on megakaryocyte differentiation. Interestingly, experiments with the immunofluorescence assay showed that the derivation of CD41a$^+$CD42b$^+$ MKs was suppressed after the incubation with the CM from three different MM cell lines including ARP1, RPMI-8226, and OPM2 (Supplementary Fig. 5d). The results from the flow cytometry assay further confirmed the inhibition of megakaryocytic differentiation by the CM from MM cells (Fig. 6d). We also found that the CM severely prevented the production of proplatelets and CD41a$^+$CD42b$^+$ PLP particles (Fig. 6e, f). Furthermore, the megakaryocytic differentiation of c-Kit$^+$ cells from mice was also suppressed by the CM produced from 5TGM1 cells (Supplementary Fig. 5e, f). Thus, MM cells exert inhibitory effects on megakaryopoiesis and thrombopoiesis, likely via the secretion of extrinsic factor(s).

We next directly assessed whether serine was indeed secreted from MM cells. We measured $^{13}C$-labeled serine by exploring $^{13}C_2$-glycine metabolic flux of ARP1 cells and found that $^{13}C$-labeled serine was gradually elevated in the CM of ARP1 cells (Fig. 6g). We then evaluate the effects of manipulating PHGDH expression on serine production, megakaryopoiesis, and thrombopoiesis by generating

APR1 myeloma cells with PHGDH ectopic expression or knockdown. As expected, overexpression of PHGDH increased the level of serine in the CM (Supplementary Fig. 5g), while the CM collected from PHGDH-overexpressing APR1 cells inhibited megakaryocytic differentiation and PLP generation ($p < 0.05$) (Fig. 6h-j). In contrast, knock-down of *PHGDH* decreased the level of serine in the CM (Supplementary Fig. 5h) and facilitated megakaryocytic differentiation and PLP derivation (Supplementary Fig. 5i-k). To further investigate whether serine from MM cells affected the expression of serine metabolism-related genes, we assessed the expression of those genes by RNA-seq analysis. Consistent with the mRNA expression profiling of serine treatment, *SLC38A1*, *GCSH*, and *PHGDH* were upregulated in cells treated with CM of ARP1 cells (Supplementary Fig. 5l).

Megakaryopoiesis and thrombopoiesis takes place largely in the BMME with myeloma cells and thus might be affected by the potential extrinsic factors produced by MM cells. To explore this, we determined the transition of serine to methionine in MKs upon the treatment of CM produced from MM cells (Fig. 6k). Intriguingly, enhanced labeling of methionine was observed in MKs treated with the CM of MM cells (Fig. 6l). Consistently, trimethylation of H3K9 was also upregulated in cells with the CM treatment (Fig. 6m). These results highlighted the functions of PHGDH in mediating the production of serine and its inhibitory effects on megakaryocytic differentiation and PLP generation.

### TPO administration lessens thrombocytopenia and suppresses MM progression

After discovering the decrease of platelets induced by excessive serine in the BM microenvironment of MM, we asked whether thrombocytopenia could be improved by the administration of TPO, a critical regulator of megakaryopoiesis and thrombopoiesis[23]. To this end, we next examined the effect of exogenous supply of TPO in vivo on thrombocytopenia caused by excessive serine (Fig. 7a). The serine concentration in the serum was elevated in mice fed with high-serine diet compared with control diet, but was reduced in mice with control diet and high-serine diet combined with TPO intervention (Fig. 7b). The numbers of platelets and MKs were also restored to higher levels in mice with TPO intervention (Fig. 7c, d). In addition, quantitative imaging analysis revealed enhanced infiltration of tumor cells and tumor burden in the high-serine diet group compared with control diet group at week 6, which was reduced in the control group and high-serine diet group after TPO administration (Fig. 7e, f). Moreover, the concentrations of serum IgG2b in the TPO-treated control group and high-serine diet group were lower than that in the control group or the high-serine diet group (Fig. 7g). Together, these results suggest that TPO administration suppresses MM progression and improves the survival of MM mice, at least partially by facilitating megakaryopoiesis and thrombopoiesis.

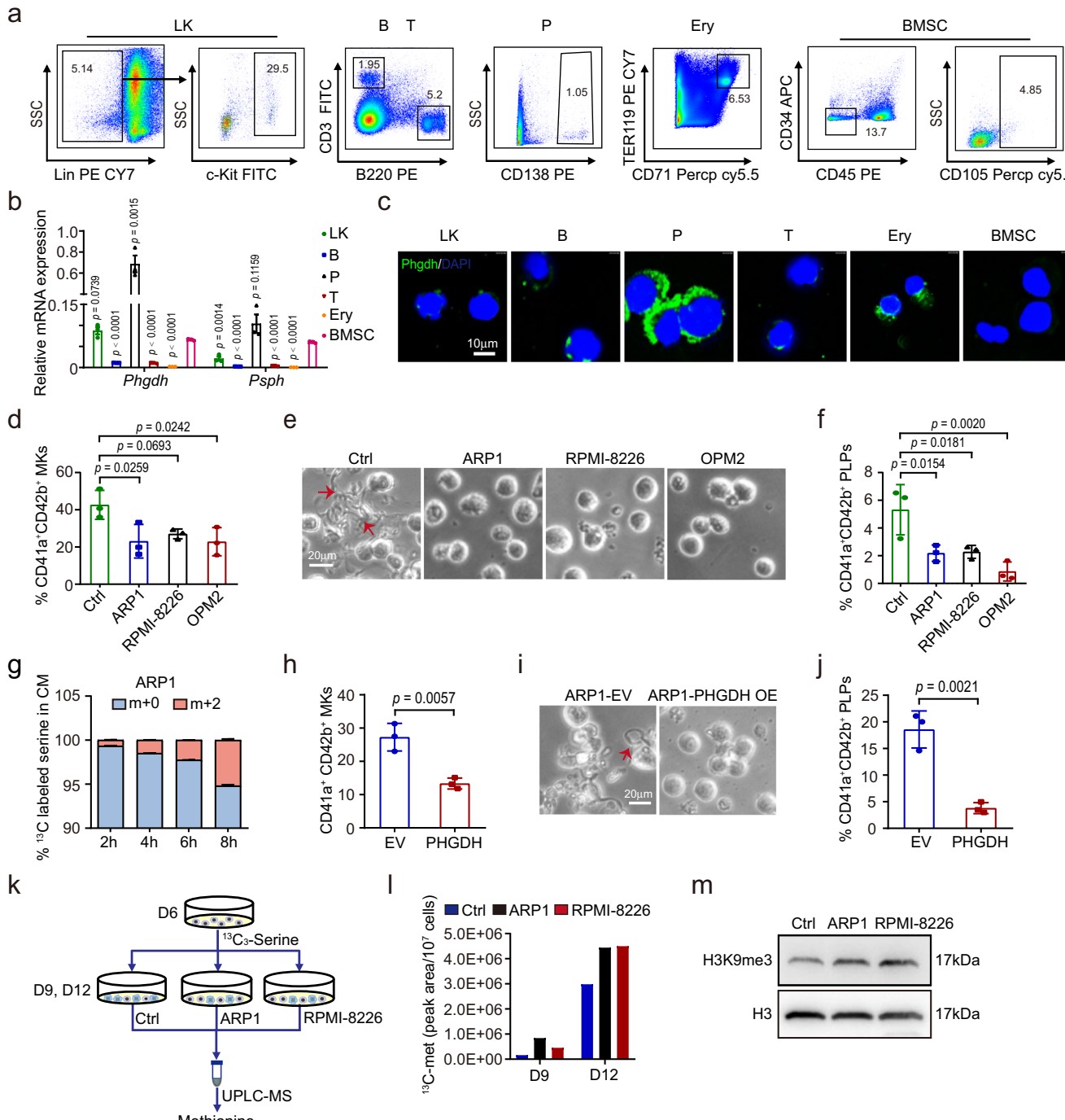

**Fig. 6 | Serine accumulated in the BM microenvironment is released from myeloma cells. a**, **b** Representative plots for the isolation of and relative mRNA levels of *Phgdh* and *Psph* were detected in LK, B cell, plasma cells, T cell, erythrocytes, and BMSCs from the BMME of 5TGM1 mice (mean ± SD, *n* = 3 independent experiments). **c** Representative images of immunofluorescence stained with Phgdh in different cells. The scale bar is 10 μm. **d** Flow cytometry analysis of the percentage of CD41a⁺CD42b⁺ cells at day 12 of megakaryocytic differentiation treated with or without the CM from MM cells (mean ± SD, *n* = 3 independent experiments). **e**, **f** Phase-contrast images of proplatelet formation and generation of CD41a⁺CD42b⁺ PLPs at day 12 from CD34⁺ cells treated with or without the CM from MM cells (mean ± SD, *n* = 3 independent experiments). The scale bar is 20 μm. **g** ¹³C-labeled serine in the CM from ARP1 cells treated with ¹³C₂ glycine (mean ± SD, *n* = 3 independent experiments). **h** Flow cytometry analysis of the percentage of

CD41a⁺CD42b⁺ MKs from CD34⁺ cells treated with CM of ARP1-EV and ARP1-PHGDH cells (mean ± SD, *n* = 3 independent experiments). **i**, **j** Representative images of proplatelet formation and generation of CD41a⁺CD42b⁺ PLPs at day 12 from CD34⁺ cells treated with CM derived from ARP1-EV and ARP1-PHGDH cells (mean ± SD, *n* = 3 independent experiments). The scale bar is 20 μm. **k** Schematics of the serine metabolic flux experiments treated with CM from MM cells. **l** Levels of ¹³C-methionine in differentiated cells after incubated with the CM of ARP1 cells and RPMI-8226 cells with ¹³C₃-serine for 4 h. **m** Western blot analysis of H3K9me3 in differentiated cells at day 12 treated with control medium or CM from ARP1 cells and RPMI-8226 cells. Experiment was repeated independently for 3 times. An unpaired two-sided Student's *t*-test was used in **b**, **h**, **j**; One-way ANOVA followed by Dunnett's multiple comparison test in **d**, **f**. Source data are provided as a Source Data file.

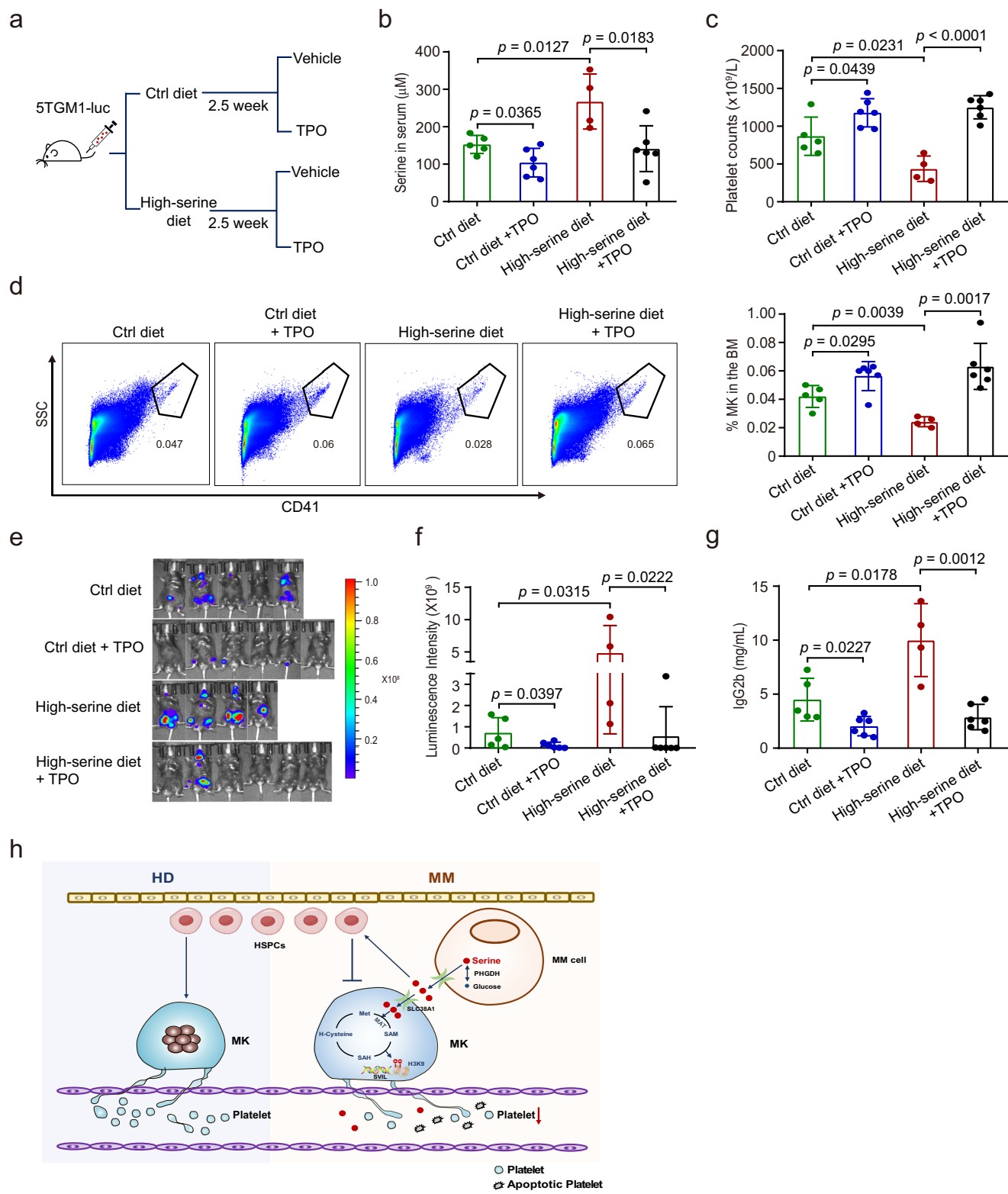

## Discussion

In the present study, we showed that thrombocytopenia correlates with poor prognosis of NDMM patients, especially for those at ISS stage III that are also accompanied with progressive loss of mature MKs. Further un-targeted and targeted metabolic assay showed that the serine level is highly elevated in MM patients with thrombocytopenia. The excessive production of serine is originated from MM cells, while the impact of serine on thrombocytopenia is mainly mediated through the suppression of MK differentiation in the BM and the induction of apoptosis of platelets in the PB. Extrinsic serine uptake by SLC38A1 inhibits SVIL expression via SAM, leading to the repression of megakaryopoiesis and thrombopoiesis (Fig. 7h). Finally, inhibition of serine utilization and the pharmacological treatment with TPO restore megakaryopoiesis and the production of platelets, suggesting a promising strategy to improve the therapeutic outcomes of MM.

**Fig. 7 | TPO administration lessens thrombocytopenia and suppresses MM progression. a** A schematic diagram of 5TGM1 mice fed with the control diet (ctrl diet), ctrl diet plus TPO intervention (1ug/mouse/day), high-serine diet or high-serine diet plus TPO intervention (1ug/mouse/day) (*n* = 6 mice per group). **b** Serum serine concentrations in 5TGM1 mice (*n* = 5 in ctrl diet group, *n* = 4 in high-serine diet group, *n* = 6 in ctrl diet plus TPO intervention, *n* = 6 in high-serine diet plus TPO intervention) at week 6. Results represent means ± SD. **c** Platelet counts in PB of 5TGM1 mice (*n* = 5 in ctrl diet group, *n* = 4 in high-serine diet group, *n* = 6 in ctrl diet plus TPO intervention, *n* = 6 in high-serine diet plus TPO intervention) at week 6. Results represent means ± SD. **d** The percentage of MKs in BM cells of 5TGM1 mice (*n* = 5 in ctrl diet group, *n* = 4 in high-serine diet group, *n* = 6 in ctrl diet plus TPO

intervention, *n* = 6 in high-serine diet plus TPO intervention) as revealed with flow cytometry analysis. Results represent means ± SD. **e, f** Live imaging and quantification of the tumor-associated luminescence intensity of the four groups of 5TGM1 mice (*n* = 5 in ctrl diet group, *n* = 4 in high-serine diet group, *n*=6 in ctrl diet plus TPO intervention, *n* = 6 in high-serine diet plus TPO intervention) at week 6. Results represent means ± SD; Significance was analyzed with an unpaired one-sided Student's *t*-test. **g** Serum IgG2b concentrations in 5TGM1 mice (*n* = 5 in ctrl diet group, *n* = 4 in high-serine diet group, *n* = 6 in ctrl diet plus TPO intervention, *n* = 6 in high-serine diet plus TPO intervention) at week 6. Results represent means ± SD. Unpaired two-sided Student's *t*-test was used in **b, c, d, g. h** The working model of the study. Source data are provided as a Source Data file.

Earlier studies demonstrated that MM patients have lower platelet levels, which correlate with poor prognosis, although the criteria of thrombocytopenia are defined differently in these studies[4,5,24,25]. However, whether thrombocytopenia can be integrated into the prognostic classification systems of MM and the correlation between thrombocytopenia and clinical pathology are still unclear. Here we systematically analyzed the role of thrombocytopenia in the pathogenesis of MM from the following aspects. First, we discovered that lower platelet numbers are associated with more severe disease outcomes. Approximately, 30.4% of the MM patients have low platelet number (≤120 × 10⁹/L) accounted. Furthermore, the patients with low platelet counts (7% of total MM patients) suffer from severe life-threatening bleeding, such as gastrointestinal hemorrhage and intracranial hemorrhage (Supplementary Table 4). The data of 1,468 patients from three centers in China showed that thrombocytopenia is much more severe than previously reported[3,26] and clearly merits further attention in the future. The advanced disease stage and the genetic variations are postulated to be responsible for the increased proportion of thrombocytopenia of MM in China. Moreover, we found that the patients with thrombocytopenia correlate with anemia and genetic heterogeneity, including 1q21 gain and *TP53* deletion, but it is unclear whether the genetic heterogeneity affects thrombocytopenia and whether myeloid progenitor cells are also affected by the microenvironment of myeloma when the progenitor cells differentiate into MKs. Third, the nutritional conditions and metabolic factors necessary for the survival of residual MK cells in BM are still poorly defined. Previously, we found that the metabolic imbalance of amino acids in BM promotes the proliferation of MM cells[13]. We also revealed the existence of heterogeneous subpopulations of MKs with distinct functions in the human embryos and BM by using single-cell RNA-seq[27,28]. Taken together, our earlier findings suggested that the heterogeneity of MKs may also exist under the pathological conditions of MM. On the one hand, the heterogeneous nature of MKs may explain why the MM patients with poor prognosis have the normal number of platelets. On the other hand, the residual genetically heterogeneous MK cells might require specific nutrients and factors to survive in the BM. But how the different subpopulations of MKs respond to the microenvironment and then exert their different regulation functions need further investigation. In the present study, we showed that the patients with high serine concentrations in the BM are linked to thrombocytopenia, suggesting that serine might serve as a survival factor for the residual MKs.

MM is characterized by the infiltration of malignant plasma cells into the BM microenvironment where MM cells interact with other cell types to promote MM cell growth and drug resistance[29,30]. Here, we revealed the decrease of MKs and MEPs in MM. Several earlier studies reported that CD41⁺CD42d⁺ MKs and CD34⁺ stem/progenitor cells are severally impaired due to their interactions with tumor cells, stromal cells or soluble cytokines, leading to the impairment of MK differentiation[31,32]. Our recent study showed that the concentrations of several amino acids are significantly altered in BMME and that the level of one of these amino acids - glycine is elevated due to bone collagen

degradation mediated by MM cell-secreted matrix metallopeptidase. MM cells utilize the channel protein coded by solute carrier family 6 member 9 (SLC6A9) to absorb extrinsic glycine, which is then involved in the synthesis of GSH and purines to contribute MM progression[12]. This earlier study implies that the imbalance of amino acids in BMME accelerates MM progression by interacting with different cell types directly or indirectly. In the current study, we observed increased level of serine in the BM plasma of MM patients with thrombocytopenia. Interestingly, our previous work had shown that PHGDH, the rate-limiting enzyme in serine biosynthesis, promotes proliferation and resistance of MM cells to BTZ by elevating GSH[14]. In the present study, we further demonstrated that PHGDH expression in MM cells is higher than in other cell types and that MM cells are the main source of serine where serine is released to the BMME. Extrinsic serine acts on MKs in the BM and induces apoptosis of platelets in the PB to induce thrombocytopenia in the MM patients. These findings revealed a pathological microenvironment in MM that influences megakaryopoiesis as well as thrombopoiesis and provide potential targets for therapeutic interventions.

Serine is involved in many critical metabolic processes, including protein biosynthesis, GSH synthesis, and one-carbon unit metabolism[33,34]. Serine is incorporated into S-adenosylmethionine through one carbon unit metabolism and provides methyl donors for the methylation of proteins and DNA in cells[35]. We found that serine participates in the methionine cycle after entering the cell, promotes the trimethylation of lysine of histone H3, leads to transcriptional inhibition, and silences the expression of target genes. Therefore, inhibiting the catabolism of one-carbon unit of serine may have potential therapeutic significance for preventing thrombocytopenia caused by excessive serine. The differentiation of HSPCs into MKs is mainly regulated by the TPO signaling pathway and transcription factors including RUNX1, FLI1, GATA1, and MEIS1, which play important roles in megakaryocyte differentiation[36-38]. In our current study, we found that serine can not only regulate the expression of RUNX1, but also down-regulate SVIL. SVIL is an actin-binding protein associated with cytokinesis[39]. Numerous actin-binding proteins such as non-muscle myosin IIA (NMIIA), α-actinin1, and cofilin1 play crucial roles in proplatelet generation[40-42]. However, the role of SVIL in megakaryocyte differentiation has not been reported previously. We found that both SVIL and RUNX1 can promote megakaryocyte differentiation and platelet production. However, in contrast to RUNX1, SVIL is regulated by histone trimethylation mediated by serine release. Whether inhibition of serine metabolism or histone trimethylation can alleviate thrombocytopenia in MM remains to be studied.

The results from our study also pave the way for potential interventions that target MM from the perspective of abnormal amino-acid metabolism and regulation to effectively improve the survival of patients with MM. Specifically, our present study explored the therapeutic potential in three aspects: the identity of the amino acid (serine), the metabolic enzymes mediating the uptake and utilization of serine, and the application of TPO that functions through its receptor MPL (myeloproliferative virus ligand). TPO is widely expressed on the

surface of HSPCs and MKs and stimulates the production and differentiation of MKs and consequently the expansion of platelets. Thus, maintaining the normal platelet levels in MM patients during the treatment may be an effective strategy in the treatment of MM in an individualized manner. Although mechanistic explanations for why elevated platelets and MKs inhibit MM progression are still lacking, we speculate that the following possibilities, alone or in combination, might account for the inhibition: First, PF4 is a major chemokine secreted from megakaryocyte which inhibits the progression of MM[43,44], suggesting that MKs may inhibit the progression of MM by secreting PF4. Second, several studies have reported that MKs play a critical role in maintaining HSC homeostasis[45]. Third, platelets and MKs serve as immune regulators, as shown by studies from us and others[27,46,47]. A more comprehensive analysis of the detailed mechanism underlying platelet and MK inhibition of MM progression awaits future experimentation.

Together, we showed that the degree of thrombocytopenia may assist in the risk stratification of MM and provide new insights into the treatment of newly diagnosed MM patients. The high level of serine may be indicative of myeloma-related thrombocytopenia and may open avenues for potential dietary and therapeutic interventions such as serine-restrictive diets and TPO application for MM patients.

## Methods

### Study approval

All animal experiments were conducted in accordance with protocols approved by the Institutional Animal Care and Use Committees of Central South University and Hunan Normal University, China (animal experimental license number: D2021016). All human BM samples were obtained with written informed consent from patients or their guardians prior to participation in the study. The experimental protocol was approved by the Institutional Review Board of Central South University, the Chinese Academy of Medical Sciences, and Peking Union Medical College.

### Patient samples

For the analysis of the correlation of platelet counts and MM progression, a total of 1,468 patients with NDMM were enrolled into the multicenter retrospective study. 1,009 patients diagnosed with MM from the Institute of Hematology and Blood Diseases Hospital, Chinese Academy of Medical Sciences (Tianjin) between 1995 and 2014, 384 from Peking Union Medical College (Beijing) between 2006 and 2017, and 75 from the first and the third Xiangya Hospital, Central South University (Changsha) between 2014 and 2019. For the analysis of the correlation of platelet counts and MKs, the records of BM smears from 16 cases of healthy donors and 98 cases of MM patients were obtained from the Institute of Hematology and Blood Diseases Hospital (Tianjin). 30 BM plasma samples for un-targeted metabonomics analysis were collected from the first and the third affiliated Xiangya Hospital, Central South University (Changsha). 114 BM plasma samples for targeted metabonomics analysis and 44 primary bone marrow biopsies of MM patients for CD41 immunostaining were from the Institute of Hematology and Blood Diseases Hospital (Tianjin).

### Cell lines, antibodies, and reagents

ARP1 and RPMI-8226 cell lines were obtained from Cancer Research Institute, Central South University. OPM2 cell line was obtained from the Institute of Hematology and Blood Diseases Hospital, Chinese Academy of Medical Sciences. Luciferase-labeled 5TGM1 cell line was kindly provided by Prof. Yuhuan Zheng (West China Center of Medical Sciences, Sichuan, China). ARP1, RPMI-8226, OPM2, and 5TGM1 cells were authenticated using short tandem repeat (STR) analysis. All cell lines were tested negative for mycoplasma and were used in our previous studies[12,14,48]. MM cells were maintained in RPMI 1640 (Hyclone GE Healthcare Life Sciences, USA) supplemented with 10% FBS (Gibco, USA), 100 units/mL of penicillin, and 100 µg/mL of streptomycin (P/S). CD34+ cells were isolated from umbilical cord blood (CB) as described previously[20]. Briefly, mononuclear selection fraction was obtained with Ficoll-Hypaque density centrifugation medium (#10771, Sigma Aldrich, USA), CD34+ cells were then purified by magnetic bead separation using the human CD34 MicroBead kit (#130-046-702, Miltenyi Biotec, Germany) as instructed by the manufacturer. CD34+ fractions showing purity higher than 95% were used for further experiments.

Anti-Human CD41a-APC (#561852), Anti-Human CD41a-FITC (#555466), anti-Human CD42b-APC (#561854), anti-mouse CD41-FITC (#561849), anti-mouse CD34-FITC (#560238), anti-mouse CD45R/B220-PE (#553089) were purchased from BD Biosciences (BD Biosciences, USA). A cocktail of antibodies to lineage committed cells (CD45R, CD19, CD11b, CD3e, TER-119, CD2, CD8, CD4, Ly-6C/G) was obtained from Miltenyi (#130-110-470, Miltenyi Biotec, Germany). PE/CY7-conjugated streptavidin (#405206), anti-mouse c-Kit-APC (#105812), anti-mouse c-Kit-PE (#105808), anti-mouse c-Kit-FITC (#161604), anti-mouse Sca-1-APC/CY7 (#108126), anti-mouse CD16/32-PE (#101308), anti-mouse TER-119-APC (#116212), anti-mouse CD71-PerCP/Cyanine5.5 (#113816), anti-mouse/human B220-PerCP/Cyanine5.5 (#103236), anti-mouse CD105-PerCP/Cyanine5.5 (#120416), anti-mouse CD3-FITC (#100204) and anti-mouse CD138-PE (#142504) were purchased from Biolegend (Biolegend, USA).

The antibodies used in this study were purchased as follows: anti-Bcl-xL (#A0209, 1: 1000), anti-SETDB2 (#A7391, 1: 1000), anti-SLC1A4 (#A12507, 1: 1000), anti-SLC1A5 (#A6981, 1: 1000), and anti-SHMT2 (#A1215, 1:1000) were all purchased from ABclonal. Anti-p53 (#10442-1-AP, 1: 1000), anti-BAD (#10435-1-AP,1: 1000), anti-SLC38A1 (#12039-1-AP, 1: 1000), anti-Histone-H3 (#17168-1-AP, 1: 5000), and anti-Beta Actin (#66009-1-Ig, 1: 5000) were purchased from Proteintech Group. Anti-Supervillin (#Sc-53556, 1: 500 for western blot, 1: 50 for immunofluorescence) was obtained from Santa Cruz. Anti-Tri-Methyl-Histone H3 (Lys9) (#13969, 1: 1000) and anti-Cleaved Capase-3 (Asp175) (#9661 S, 1: 1000) were obtained from Cell Signaling Technology.

### 5TGM1 mice model

C57Bl/KaLwRijHsd mice of 6 weeks old were purchased from Harlan Laboratories Inc. (Harlan Netherlands BV, The Netherlands). All mice were maintained under SPF conditions in a controlled environment of 20–22 °C, with a 12/12 h light/dark cycle, 50–70% humidity. 6-8 weeks-old male or female mice were inoculated intravenously (IV) with phosphate-buffered saline vehicle (Ctrl mice) or $1 \times 10^6$ luciferase labeled 5TGM1 cells (5TGM1 mice). In general, 5TGM1 mice reaching endpoint criteria when they become sick (symptoms include weight loss, appetite loss, hunched posture, leg lame, or bone fracture), then mice were euthanized by intraperitoneal injection of 150 mg/kg sodium pentobarbital. The maximum approved serum IgG2b concentration is 20 mg/mL and tumor-associated luminescence intensity is $1.04 \times 10e10$ in our study, which was permitted by the Institutional Animal Care and Local Veterinary Office and Ethics Committee of the Central South University. For the analysis of dynamic changes of MKs, MEPs, CMPs, and LSK cells, healthy and myeloma-bearing mice were sacrificed in weeks 4 and 6 post-inoculation. The progression of the tumors was assessed by using serum analysis of the myeloma-specific IgG2b and bioluminescence imaging (BLI). Briefly, mice were injected intraperitoneally with 15 mg/mL luciferin in PBS at a dose of 150 mg luciferin per kilogram mouse and imaged in weeks 2, 4, 6 or 8 via BLI, followed by an incubation period of 5 minutes. The fluorescence intensity of BLI output was acquired by subtracting the background and analyzed using Living Image Software (PerkinElmer, USA). Control diet, serine-free diet and high-serine diet were prepared by a company (#TP20210609001 and #TP20220701005, Trophic Animal Feed High-Tech Company, Jiangsu, China). For the diet used in serine-free group,

the control diet was composed of essential and non-essential amino acids (1.33% serine and 16.48% other amino acids), corn starch (41.45%), dextrin and sucrose (25%), cellulose (5%), soybean oil (5%), minerals and vitamins (4.75%), tert-butylhydroquinone (0.001%), and other components. The serine-free diet was the same as the control diet, but without serine, and corn starch (42.78%) were increased proportionally. For the diet used in high-serine group, the control diet was composed of essential and non-essential amino acids (0.35% serine, 17.46% other amino acids), corn starch (42.43%), dextrin and sucrose (25%), cellulose (5%), soybean oil (5%), minerals and vitamins (4.75%), tert-butylhydroquinone (0.001%), and other components. The high-serine diet was the same as the control diet, but with 2.8% serine, and corn starch (39.98%) were decreased proportionally. In the rhTPO intervention study, rhTPO (Peprotech, USA) was injected subcutaneously (SC) from week 2.5 post tumor inoculation for 4 consecutive days (1 μg/mouse/d)[49].

To assay platelet counts, fifty microliters of blood samples were collected and transferred to EDTA-containing bottles from the 5TGM1 mice, and complete blood counts were measured with Mindray BC-5100 hematology analyzer (Jinan, China). PB serum was obtained by the centrifugation of blood samples at 10,000 g for 10 min and stored immediately at −80 °C. The serum concentration of IgG2b was determined by using enzyme-linked immunosorbent assay (ELISA) (Mouse IgG2b ELISA Quantitation Set, Bethyl Laboratories) according to the manufacturer's instructions.

## MK differentiation

Megakaryocytic differentiation was performed and in vitro platelets/PLPs was detected as previously described[20,39]. CD34[+] cells were cultured in StemSpan™ SFEM (StemCell Technologies, Canada) supplemented with 1% P/S (Gibco, USA) and cytokines hTPO (50 ng/ml, Peprotech), hIL-3 (20 ng/ml, Peprotech), hSCF (20 ng/ml, Peprotech). After 6 days of expansion, cells were transferred to StemSpan™ SFEM supplemented with hTPO (50 ng/mL) and hIL-11 (20 ng/mL) for megakaryocytic differentiation. For cells treated with CM, fresh medium was mixed at a 1:1 ratio with the CM for MK differentiation. For cells treated with amino acids, amino acids were added directly to each well at day 6 of MK differentiation. For MK differentiation of c-Kit[+] cells from the control mice, cells were cultured in the presence of 50 ng/mL murine TPO (mTPO, Peprotech) and 20 ng/mL murine SCF to obtain murine MKs.

For the collection of the CM from 9 MM cell lines, named ARP1 CM, OPM2 CM, RPMI-8226 CM, 5TGM1 CM, ARP1-EV CM, ARP1-PHGDH CM, ARP1-Scramble-CM, ARP1-PHGDH-sh1 CM, ARP1-PHGDH-sh2 CM, respectively. ARP1, OPM2, RPMI-8226, ARP1-EV, and ARP1-PHGDH cells (4×10[4]/mL) were plated onto 60-mm culture dishes in StemSpan™ SFEM (StemCell Technologies, Canada). Culture supernatants were harvested 96 hours later, and after cell debris was removed by centrifugation, cells were stored at -80 °C. To obtain the CM of ARP1 cells with PHGDH knockdown, 2 ng/mL dox was applied to ARP1-PHGDH-sh1 and ARP1-PHGDH-sh2 cells. CM were harvested 24 hours later and then added to the MK differentiation system.

## Flow Cytometry Analysis

For the analysis of the proportion of MKs in vitro, cells were collected by centrifugation, resuspend with 0.1% BSA containing CD41a-APC, CD42b-PE antibodies, Hoechst 33342, and then incubated in dark at room temperature (RT) for 30 min before flow cytometry analysis. For PLP analysis, culture supernatant was incubated with CD41a-APC and CD42b-PE antibodies at RT for 30 minutes. Mouse IgG1κ APC/PE-conjugated Isotype Control antibody (BD bioscience, USA) was used as a negative control.

For the analysis of MKs in 5TGM1 mice, BM cells were collected by flushing the femurs and tibias with PBS. Bone marrow cell suspensions were filtered (100-μm filter) and used for the analysis of the proportion

of CD138[+] MM cells, MKs (SSC[high]CD41[high]), LSK (Lin[−]/Sca-1[+]/c-Kit[+]), CMPs (Lin[−]/Sca-1[−]/c-Kit[+]/CD34[+]/CD16/32[−]) and MEPs (Lin[−]/Sca-1[−]/c-Kit[+]/CD34[−]/CD16/32[−]), by using a BD FACS CantoII or BD LSRII Flow cytometer (BD Biosciences, CA, USA).

For the analysis of apoptotic cells or platelets, cells were labeled by FITC-conjugated Annexin V according to the manufacturer's instructions (US Everbright Inc, USA). Cells were then analyzed on an LSR II cytometer (BD Biosciences, USA), and the results were analyzed using FlowJo V10 software (BD Biosciences, USA). Each test was repeated three times.

## Immunofluorescence

For the staining of MKs, the femurs from control mice and 5TGM1 mice were fixed in 4% paraformaldehyde solution, decalcified in 0.5 M EDTA for 24 hours at 4 °C, embedded with Tissue-Tek O.C.T Compound (OCT) and store at −80 °C, and then sectioned to a thickness of 30 μm. The frozen-sections were hydrated with PBS for 5 min and permeabilised in 0.3% TritonX-100 for 15 min and were then blocked through immersion in 2% BSA (diluted by PBS) for 30 minutes. CD41-FITC (1:100 dilution), SVIL (1:50 dilution), and PHGDH (1:50 dilution) were incubated with the sections overnight at 4 °C. Nuclei were stained with 2 μg/mL Hoechst 33342 or DAPI (BD bioscience, USA), and the images were photographed on a confocal microscope (PerkinElmer, USA). For the staining of cells from megakaryocytic differentiation, cells were incubated with CD41-FITC (1:100 dilution) and CD42-APC (1:100 dilution) overnight at 4 °C, Hoechst 33342 (2 μg/mL) was added directly to the cells, and incubated at RT for 30 minutes before imaging.

## High-performance liquid chromatography (HPLC)

CD34[+] cells were expanded for 6 days and then induced to undergo megakaryocytic differentiation in fresh medium supplemented with or without serine (8 mM). Cells were washed twice with ice-cold phosphate-buffered saline, resuspended in 80 μL of 10 mM HCl, and then lysed via two freeze/thaw cycles, followed by the addition of 20 μL of 5% 5-Sulfosalicylic acid (5-SSA). Cell debris was removed by centrifugation. For serum samples, 10 μL serum was added to 20 μL of 1% 5-SSA and placed on ice for 10 min, and the precipitated proteins were removed by centrifugation (15,000 r.p.m. for 10 mins at 4 °C). Supernatants were analyzed on an LC platform consisting of an LC-20A detection system (SHIMADZU, Japan). The LC method employed a 18 C column (2.1 mm × 150 mm, 5 μm) (Agilent, USA) with the mobile phase mixed by A = sodium acetate 50 mM (adjusted to pH 6.8) and B = methanol for metabolite separation. The total run time was 40 min.

## Western blotting

The total protein of cultured cells was extracted with RIPA lysis buffer from CWBIO (ComWin Biotech Co., Ltd, China). After the detection of protein concentrations of each sample, equal amounts of protein extracts were separated by using 8–12% SDS-PAGE and then transferred to a polyvinylidene fluoride membrane (Millipore-Sigma, USA). The membrane was blocked with 5% milk nonfat dry milk and incubated with primary antibodies at 4 °C overnight, followed by secondary antibodies conjugated to horseradish peroxidase. The immunoreactivity was detected with the Super Signal West Pico Chemiluminescent Substrate or the West Femto Trial kit (Thermo Fisher Scientific, USA).

## Vectors, transfections, and transductions

Short hairpin RNA (shRNA) sequences targeting human *SLC38A1*, *SHMT2, MAT2A, SVIL*, and *SHANK3* were annealed and ligated into a pLKO-GFP lentiviral vector. Lentiviruses were packaged in HEK293T cells using pMD2G and psPAX2 helper vectors and polybrene (8 μg/mL)-mediated transduction. The supernatant of lentiviruses was centrifugated (20,000 rcf for 2 hours at 4 °C). CD34[+] cells transduced with recombinant lentivirus (GFP[+] positive cells) were selected for

further analysis. All shRNA sequences are listed in Supplementary Table 5.

## Quantitative real-PCR

Total RNA was isolated using Trizol reagent and reverse transcribed using the SuperScript™ II Reverse Transcriptase kit (Thermo Fisher Scientific, USA) following manufacturer's protocol. Real-time quantitative PCRs (qPCR) were performed by using ABsolute qPCR SYBR Green Mixes (Thermo Fisher Scientific, USA). To calculate the Fold changes of the indicated genes, the comparative CT method ($2^{-\Delta\Delta Ct}$) was used, and β-actin was the loading control. Primer sequences are provided in Supplementary Table 6.

## Chromatin Immunoprecipitation

The binding of the *SVIL*, *SHANK3*, and *FRYL* promoter regions to H3K9me3 in 293T treated with vehicle or serine was quantified with ChIP-quantitative PCR (ChIP-qPCR). The chromatin immunoprecipitation (ChIP) assay was performed with the EZ-ChIP kit (Millipore, USA), Briefly, chromatin from the 293T cells treated with vehicle or serine was used in the ChIP assays using antibodies against H3K9me3 (Cell Signaling Technology, USA). Normal IgG was used as negative control. ChIP-qPCR was used to amplify various promoter regions of the target gene, *GAPDH* was amplified with the corresponding primers as a negative control. The primers are listed in Supplementary Table 7.

## Metabolomics analysis

Gas chromatography coupled to time-of-flight mass spectrometry (GC-TOFMS) system (Pegasus HT, Leco Corp., St. Joseph, USA) was used to quantify the detected metabolites (Metabo-Profile Biotechnology Co. Ltd, China)[50]. The experimental method was as previously described[13]. Briefly, bucket tables were imported into SIMCA-P 14.0 software (Umetrics AB). Orthogonal partial least squares discriminant analysis (OPLS-DA) was conducted to produce models of "best fit" between the groups of samples. The variable importance in projection (VIP) >1 and fold change (FC) > 1.2 were considered as differential metabolites. Liquid chromatography tandem-mass spectrometry (LC-MS) was used for quantitative analysis of amino acids in the BM plasma from MM patients with normal platelet counts or thrombocytopenia.

## $^{13}C_3$-serine and $^{13}C_2$-glycine labeling experiments

CD34$^+$ cells were plated in 10-cm dishes in StemSpan™ SFEM medium. After 6 days of expansion and 3 or 6 days after megakaryocytic differentiation, cells were counted and then cultured in fresh medium supplemented with 100 ng/mL $^{13}C_3$-serine (Yifei Biological Technology Co., Ltd, China) for 2 h before metabolite extraction[51]. The time of addition of tracer media was designated as time 0. To mimic the tumor microenvironment of MM, CD34$^+$ cells were cultured with the CM of ARP1 and RPMI-8226 cells on day 6 and subsequently transferred to the CM supplemented with 100 ng/mL $^{13}C_3$-serine on day 9 and day 12 of megakaryocyte differentiation. Cells were washed three times with cold PBS after 4 h of tracing, and the dry pellets were stored at −80 °C. For analysis the release of serine from MM cells, ARP1 cells cultured in fresh medium supplemented with 10 ng/mL $^{13}C_2$-glycine for 2 h, 4 h, 6 h, and 8 h before supernatant collection. All samples were used for liquid chromatography triple-quadrupole mass spectrometry (UPLC-TQ-MS) analysis (Metabo-Profile Biotechnology Co. Ltd, China).

## ATAC-seq (assay for transposase-accessible chromatin using sequencing)

ATAC-seq analysis was performed in differentiated cells treated with or without serine at day 12 of megakaryocytic differentiation (Novogene Biotechnology Co. Ltd, China). First, Peaks were called ($p$ <0.05) with macs2 software. Next, read counts, the RPM value, and FoldEnrich value of the merged peaks were calculated, the Log$_2$FC value was then calculated by using the FoldEnrich value of the peaks of the two

groups. Differentially accessible peaks were subsequently screened out and identified with the criterion of log$_2$FC > 1. Differentially accessible peaks were then ascertained using the chip-seeker software. Finally, the distance between the peak and the transcription start site (TSS) of the gene was calculated, and the gene with the nearest TSS distance was considered as the peak-related gene. ATAC-seq reads were mapped to the hg38 reference genome using bowtie[52]. Peaks were displayed using the Integrative Genomics Viewer software.

## RNA-seq analysis

RNA-seq analysis was performed in differentiated cells with various treatments at day 12 of megakaryocytic differentiation, including cells treated with ARP1 CM or with serine. Total RNA was isolated using Trizol reagent following manufacturer's protocol. The mRNA is fragmented into short fragments and cDNA is synthesized using the mRNA fragments as templates. The short fragments were then purified and connected with adapters, and the transcriptome data of those cells were profiled by using a BGISEQ-500 (Beijing Genomics Institute at Wuhan, China). After quality control (QC) analysis was conducted, the read counts were generated by aligning the human genome assembly. After the removal of improperly aligned reads, FPKM (Reads Per Kilobase of exon model per Million mapped reads) values were generated by StringTie with a provided Python script. Differentially expressed genes were identified if they had a FC > 1.2 between the two groups.

## Statistics & Reproducibility

Relations between platelet counts and discrete variables were tested by using the two-sided chi-square. Kaplan-Meier survival curves were evaluated by using a two-sided log-rank test. Data were analyzed using a student unpaired *t*-test for the comparison of two groups, and one-way analysis of variance (ANOVA) or multiple *t* tests for the comparison of more than two groups. $p$ <0.05 was considered statistically significant. No data were excluded from the analyses.

## Reporting summary

Further information on research design is available in the Nature Portfolio Reporting Summary linked to this article.

## Data availability

The raw RNA-sequencing and ATAC-seq data generated in this study have been deposited in the public database of Genome Sequence Archive (GSA) for Human under the accession number HRA003851 and HRA003852, which are accessible at and, respectively. The untargeted metabolomics data and targeted metabolomics data were included in an Excel form which is named untargeted metabolomic data and targeted metabolomic data in the Source Data file. Source data are provided with this paper as a Source Data file. The remaining data are available within the Article, Supplementary Information, or Source Data file. Source data are provided with this paper.

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

## Acknowledgements

The authors thank Wenying Yu, Weichao Fu, Ting Chen, and Wanzhu Yang for technical supports with flow cytometry and confocal microscope. We also thank professors Rushi Liu and Kaiqun Ren from Hunan Normal University School of Medicine for their assistance with bioluminescence imaging. Thanks to the Animal Center of Hunan Normal University School of Medicine for providing the experimental platform. This work was supported by the National Natural Science Foundation of China (82130006, 81974010 to W. Zhou, 82125003, 81870099 to J. Zhou, 82200231 to C. Kuang); Haihe Laboratory of Cell Ecosystem Innovation Fund (22HHXBSS00030 to W. Zhou, 22HHXBSS00031 to J. Zhou); CAMS Innovation Fund for Medical Sciences (CIFMS) (2021-I2M-1-073, 2022-I2M-JB-015 to J. Zhou); The Ministry of Science and Technology of China (2018YFA0107800 to W. Zhou); Natural Science Foundation of Hunan Province (2020WK2006 to W. Zhou); Strategic Priority Research Program of Central South University (ZLXD2017004 to W. Zhou); SKLEH-Pilot Research Grant (ZK21-05, ZK22-06 to W. Zhou); China Postdoctoral Science Foundation (2021M703631 to C. Kuang); Postdoctoral Science Foundation of Central South University (140050025 to C. Kuang); Postgraduate Research Innovation Project of Hunan Province, China (CX20220348 to C. Hu); Fundamental Research Fund for Graduate of Central South University (2022ZZTS0266 to C. Hu).

## Author contributions

Contribution: W.Z. and J.Z. conceived of and designed the experiments; C.K., M.X., C.H., Jingyu Zhang, B.M., J.X., X.W., J.G., and Y.Z. performed the experiments; C.K., M.X., C.L., C.H., Jingyu Zhang, B.M., X.L., Z.L., X.W., and X.F. analyzed the data; G.A., P.S., L.Q., J.L., Y.S., and Y.H. provided critical materials; W.Z., J.Z., and C.K. wrote the manuscript and all authors edited the manuscript.

## Competing interests

The authors declare no competing interests.
