## [Peer Review File · Nature Communications]

Excessive serine from the bone marrow microenvironment impairs megakaryopoiesis and thrombopoiesis in Multiple MyelomaReviewers' Comments:

Reviewer #1:

Remarks to the Author:

In this paper, authors demonstrate that thrombocytopenia in MM is induced by serine production of MM cells. Blocking serine uptake enhances thrombopoiesis and suppresses MM progression. This project is novel and interesting and the methodology is sound.

Some questions/comments remain:

- 1) For the MK differentiation with amino acids, was the standard culturing medium for these cells first depleted of amino acids or were they added in excess?
- 2) Since serine starvation has also been shown to impact MM cell survival (DOI: 10.1186/s40170-017-0169-9), how can you be sure that what you see with the serine poor diet is not a result of direct cytotoxic effects on the MM cells (fig 3)? (Less MM cells would also lead to less thrombopenia).
- 3) Regarding fig 5 c-e, did authors also try to add methionine to MK with SLC38A1-sh? This would indicate that methionine triggers the same effects as serine.
- 4) It would be interesting to see whether treating mice with a PHGDH inhibitor would lead to 1) less serine in BM plasma and 2) more MK differentiation and less cytopenia.
- 5) The findings regarding the use of TPO are somewhat surprising since a recent paper demonstrates the opposite: <https://doi.org/10.1016/j.ajpath.2020.12.016>. In this paper, authors demonstrate that TPO is upregulated during MM progression and is negatively correlated to platelet count. Moreover, this TPO was found to be pro-tumoral. Can the authors comment on this? It is also not clear why TPO treatment would result in less serine in the serum when used during treatment with a high serine diet (fig 7b). What cells take up the serine because of the TPO treatment? Why was there no group with only TPO treatment?

Minor comment:

Typo: p 12 first line, this should not read a reduction in serum levels but rather an increase in serum levels.

Reviewer #2:

Remarks to the Author:

This is an interesting study that describes novel mechanisms that play a role in thrombocytopenia in multiple myeloma patients by analyzing blood and bone marrow from a mouse model and from patients. The paper shows that thrombocytopenia in MM patients correlates with a poor prognosis. Excessive serine, which is released from MM cells into the bone marrow microenvironment, is identified as the main inducer of thrombocytopenia through suppression of thrombopoiesis and induction of platelet apoptosis in the peripheral blood. Mechanistically, serine is shown to downregulate the actin-binding protein Supervillin via S-adenosyl-methionine (SAM)-mediated trimethylation of H3K9, leading to impaired thrombopoiesis. Finally, the authors showed that TPO administration or serine deficient diet might restore the physiological thrombopoiesis and improve survival in a MM mice model.

The authors present well-controlled experiments, and sound methodology and illustrate their findings extensively. Data is novel and improves the understanding of the pathophysiology of thrombocytopenia in this disease. However, I have a couple of general comments for re-consideration by the authors and also some specific comments on areas that require clarification and/or additions.

- In the first part of the paper, when assessing clinical and biological variables, platelet counts below

100x10⁹/L were considered for the "low platelets" group. While in other analyses an upper limit of 120x10⁹/L in the "low platelets" group was adopted. Why this discrepancy?

- Were treated-patients excluded from the cohort?
- How were MKs quantified in BM aspirates? A H&E image is included in Fig S1F but absolute numbers are presented in Fig 1e-f. Immunostaining of sections of bone marrow from patients would be more appropriate.
- How were in vitro platelets/PLP quantified? This is a critical point.
- Is serine affecting MK differentiation or specifically platelet release? In the MM mouse model, it seems that also hematopoietic progenitors are affected by disease progression. How specific is this mechanism for megakaryocytes? Does serine affect also MK ploidy?
- Serine-induced apoptosis in platelets. Is there a similar effect on cultured megakaryocytes?
- Figure 3e-f: Luminescence intensity and relative quantification are quite different. In the serine-deficient diet condition, it seems that all the mice display a very low signal in the histogram, while 50% of mice have substantial luminescence in the photographs. Why is this the case?
- Figure 3B: Was proplatelet formation analyzed on day 15 (figure label) or day 12 (figure legend)?
- Is SLC38A mainly expressed on MKs? What about hematopoietic progenitors? Is the actin-binding protein Supervillin mainly expressed by MKs? In the early or late phases of thrombopoiesis?
- The origin of serine from MM cells is another critical aspect. Non-hematopoietic cellular components of the BM were not considered. Are levels of PHGDH and PSPH significantly different in MM cells among patients with low/normal platelet counts?
- The last experimental point in the MM mice model is at week 6 throughout the manuscript. However, the effects of TPO administration were recorded until week 8 (Fig 7C). Can the authors explain the experiment design?

Reviewer #3:

Remarks to the Author:

The decrease in platelet counts is associated with poor prognosis in patients with multiple myeloma (MM). Kuang, Xia, and colleagues report that increased serine levels prevent megakaryopoiesis and thrombopoiesis and eventually cause thrombocytopenia. The influence of high serine concentrations on thrombocytopenia is mediated via the repression of megakaryocyte (MK) differentiation in the bone marrow and the induction of apoptosis of platelets in the peripheral blood (PB). The authors claim that the serine molecules are transported into MKs via SLC38A1 and decrease the actin-binding protein Supervillin (SVIL) via SAM-mediated tri-methylation of H3K9, resulting in the inhibition of megakaryopoiesis and thrombopoiesis.

This is an exciting study, but the current data are not sufficient to support the claim made by the authors.

1) This study is unfortunately unclear. On the one hand, the authors claim that serine enters the cells through the upregulation of SLC38A1, which is presumed to be important for regulating MK differentiation and thrombopoiesis. On the other hand, PHGDH is upregulated to promote serine synthesis and supports the repression of MK differentiation and thrombopoiesis. It is difficult to understand which pathway is essential here. Is it serine uptake or serine synthesis through PHGDH?

2) SLC38A1 is not the canonical transporter of serine. So why is SLC1A4/5 not involved? Do SLC1A4/5 knockdowns increase MK differentiation and H3K9me?

3) The authors did not demonstrate that serine, through feeding SAM synthesis, leads to H3K9me₃. This is an important part of this project. Once serine enters the cells, it can be used by the serine hydroxymethyltransferase (SHMT1/2) to support 1C metabolism and nucleotide synthesis. For example, is H3K9me₃ demethylated in response to SHMT inhibition? Does SHMT1/2 inhibition lead to the increase of megakaryopoiesis and thrombopoiesis and the inhibition of MM progression?

4) It is unlikely that SAM synthesis, downstream of serine catabolism, can solely explain the regulation of H3K9me3. Other histone and RNA methylation marks should also be affected. Other histone marks such as H3K36me are dynamically methylated in response to SAM abundance. Therefore, the authors should assess H3K36me and other histone marks in response to serine availability.

6) If the effects are SAM-dependent, then methionine starvation or direct MAT2A inhibition should recapitulate the inhibition of serine utilization.

7) The concentrations of serine used in cell-cultured are extreme (8 mM). Still, the authors do not show SAM levels in response to serine buildup. Therefore, the authors should trace ¹³C₃-serine into newly synthesized SAM.

Reviewer #4:

Remarks to the Author:

The work by Zhou and colleagues identifies a molecular mechanism driven by excessive levels of serum serine from multiple myeloma (MM) cells in the bone marrow (BM) niche that leads to thrombocytopenia and suppression of megakaryocyte (MK) differentiation via repression of Supervillin (SVIL) expression. This impacts MM progression and survival in vivo, and the authors further demonstrate that thrombocytopenia is linked to poor prognosis in MM patients and that administration of TPO enhances megakaryopoiesis and thrombopoiesis.

Overall, this was a very thorough study with extensive experimentation and a well-constructed manuscript. This work provides a unique and important molecular mechanism in MM that contributes to co-morbidities that the authors have demonstrated impact overall survival. I have several points that need to be addressed by the authors.

Major points:

1-Why are ISS I and ISS II groups in Figures 1c-d combined? Would it not be more appropriate to split them into separate figures?

2-The authors show that serum from MM patients leads to higher platelet apoptosis versus serum from healthy donors (Figure 3j). However, serum serine seems to vary greatly in MM patients in Fig 3k. It would therefore strengthen the authors argument if they can show differences in platelet apoptosis using MM patient serums with widely different serine concentrations.

3-Regarding the RNA-seq data in MK cells, how many replicates were performed for serine vs vehicle treatment groups? What was the fold change differences and p-values for SLC38A1, MTHFD1L, GCSH, PHGDH and the 15 genes that were also identified using ATAC-seq data? Did SLC38A1 protein levels increase as well? The authors also denote a p-value cutoff of 0.05, but do not denote if multiple testing correction was performed on the differentially expressed genes from RNA-seq. If no correction was performed, the authors should try to verify differential expression of relevant genes they follow-up on or describe via RT-qPCR considering the higher false positive rate.

4-For the ATAC-seq data (Figure 5i), how many replicates were performed for each group? How was the ATAC-seq integrated with RNA-seq? What constituted a positive result? How were ATAC-seq peaks identified and how were differentially accessible peaks subsequently identified? How were ATAC-seq peaks associated with genes? Did the authors solely focus on promoters? Is there any correlation between ATAC-seq signal alterations and gene expression changes at promoters? The authors should provide more detail of their data integration and provide some evidence for its utility in identifying genes of interest. Were peaks and differentially accessible peaks identified at the SVIL, FRYL and

SHANK3 promoters? Was the p-value cutoff corrected for multiple testing? A multiple testing correction is appropriate given the number of loci that are likely examined.

5-Related to point #4 above, the image in Figure 5i is very difficult to discern. The Y-axis for the ATAC-seq data is not labeled to show the read count range used and to further ensure that the range is consistent between treatment groups. It is also difficult to note the quality of the ATAC-seq data at such a narrow window. The authors should present the ATAC-seq data using a much larger window (~25-50kb) and include the location of genes. The authors should further denote in the figure which ATAC-seq signals/genomic regions were identified as significant peaks/sites of open chromatin and which were differentially accessible between treatment groups. I would suspect that promoters with significant increases in H3K9me3 to exhibit significant differential chromatin accessibility.

6-In regard to point #5 above, the authors should better explain how the H3K9me3 RT-qPCR results relate to the ATAC-seq image above. What do the colors (purple, gray and black) mean? Why were specific regions tested and not others (i.e. 3 regions for SVIL but only one for FRYL and SHANK3)?

7-The authors note that RUNX1 was repressed via RNA-seq but not ATAC-seq. Did the authors focus solely on the RUNX1 promoter region? Differentially expressed genes sometimes do not exhibit changes in promoter ATAC-seq signal. Were there any ATAC-seq peaks near RUNX1 that exhibited differential accessibility? Is there any evidence for 3D looping of these peaks to the RUNX1 promoter using publicly available HiC/HiChIP data? I think it would be worthwhile to investigate this given the importance of RUNX1.

8-Does serine treatment lead to decreased SVIL protein levels?

9-Similar to point #8 above, although the authors show that plasma cells express the highest levels of PHGDH (Fig 6B), the authors should demonstrate that this holds true at the protein level.

10-The authors have shown how MM cells inhibit megakaryopoiesis and thrombopoiesis. However, they also demonstrate that TPO suppresses MM progression and improves survival in MM mice. Regarding these experiments, can the authors speculate how TPO reduces serum serine levels? Can the authors further speculate how increased megakaryopoiesis and thrombopoiesis inhibits MM progression?

Minor points:

1-The authors need to be mindful of how they name genes when referring to human genes vs murine genes. Moreover, gene names need to be italicized to differentiate them from protein names.

2-On line 28 should thrombocytopenia be changed to thrombopoiesis?

3- The authors need to define GSH on line 70.

4-On lines 153-154, the authors state "the number of MKs was significantly lower in 5TGM1 mice with normal platelet counts than those with low platelet counts." Is this correct? Should it not be the opposite?

5- On line 162, "is" should be deleted.

6- In Figure 1f, was there a significant difference in the number of MK cells in the BM between MM cells with low vs a normal number of platelets?

7-What is the meaning of m+0, m+1, m+2, m+3 in Figure 5a?

8-In Fig 7a, vehicle is misspelled.

9-Line 434, a space is required between "including" and "RUNX1".

RESPONSE TO REVIEWERS' COMMENTS

Reviewer #1 (Remarks to the Author); expert in multiple myeloma:

In this paper, authors demonstrate that thrombocytopenia in MM is induced by serine production of MM cells. Blocking serine uptake enhances thrombopoiesis and suppresses MM progression. This project is novel and interesting and the methodology is sound.

Major comments

Comment 1: For the MK differentiation with amino acids, was the standard culturing medium for these cells first depleted of amino acids or were they added in excess?

Response to comment 1: We appreciate your valuable comments. There are two methods to investigate MK differentiation with amino acids - by adding or depleting related amino acids in culturing medium. Previously, we explored the function of glycine in BMME by depleting it in RPMI 1640 which can be commercial customized from BOSTER, Wuhan, China ¹. However, in this study, it is not convenient to customize the StemSpan™ SFEM culture medium specialized for the MK differentiation assay from StemCell Technologies, Canada. Therefore, in this study, we performed the assay by adding the amino acids in excess.

Comment 2: Since serine starvation has also been shown to impact MM cell survival (DOI: 10.1186/s40170-017-0169-9), how can you be sure that what you see with the serine poor diet is not a result of direct cytotoxic effects on the MM cells (fig 3)? (Less MM cells would also lead to less thrombopenia).

Response to comment 2: Thanks for your constructive suggestions. We draw the conclusion mainly from the following three aspects based on clinical relevance and functional studies between serine and thrombopenia. **First**, we observed that MM patients with high stage are more prone to develop thrombocytopenia. However, we observed that serine level in BMME is not significantly changed in MM patients with low platelets in ISS stages I and II, but is highly altered in MM patients with low platelets in ISS stage III, suggesting that tumor burden is not the only reason for thrombocytopenia (Fig. R1a). **Second**, the starvation of amino acids does not always delay MM progression by increasing platelets. In our study, it was found that serine and glycine were elevated in MM patients with thrombocytopenia (Fig. 2e-h in the revised manuscript). To investigate which amino acid(s) has an impact on MM development or thrombocytopenia, we applied MK differentiation assay and 5TGM1 mice model with or without serine and glycine *in vitro* and *in vivo* (Fig. R1b). Glycine was utilized by MM cells via glycine transporter SLC6A9 and accelerated the growth of MM cells¹, while glycine showed minimal effects on megakaryopoiesis and thrombopoiesis. In contrast, serine accumulation exhibits inhibitory effects on megakaryopoiesis and thrombopoiesis (Fig. R1c, d, Supplementary Fig. 2b-d). Intriguingly, we found that inhibition of glycine utilization delays the progression of MM, while showing minimal effects on platelet count in 5TGM1 mice at an early stage (Fig. R1e, f). Serine depletion not only delays the progression of MM, but also leads to reduced thrombocytopenia at an early stage (Fig. R1g, h, Fig. 3e, h in the revised manuscript). These results indicate that serine deprivation delays MM progression accompanied by less thrombocytopenia. **Third**, TPO is a key factor that facilitates megakaryopoiesis and platelet production without affecting the growth of MM cells². TPO intervention not only improves platelet counts (Fig. R1i becomes Fig. 7c, new data), but also inhibits MM progression in 5TGM1 mice (Fig. R1j becomes Fig. 7e, new data), which further suggests that increasing the number of megakaryocytes and platelets inhibits tumor progression. Therefore, we propose that the fact that the serine-free diet leads to reduced thrombocytopenia is not a result of direct cytotoxic effects on the MM cells.

Figure R1. High serine contributes to MM progression and thrombocytopenia

a Serine level in BMME of NDMM patients with normal platelet counts ($120\text{-}300 \times 10^9/\text{L}$) and low platelet counts ($\leq 120 \times 10^9/\text{L}$) at different ISS stages. **b** Schematics of the $\text{CD}34^+$ cells undergoing MK differentiation treated with vehicle or serine. **c** Generation of $\text{CD}41\text{a}^+\text{CD}42\text{b}^+$ MKs of $\text{CD}34^+$ cells treated with vehicle, serine and glycine. **d** Generation of $\text{CD}41\text{a}^+\text{CD}42\text{b}^+$ PLPs of $\text{CD}34^+$ cells treated with vehicle, serine and glycine. **e** Live imaging of the tumor-associated luminescence intensity of 5TGM1 mice fed with the control diet or glycine-free diet at week 6. **f** Platelet counts in PB of 5TGM1 mice fed with the control diet or glycine-free diet. **g** Live imaging of the tumor-associated luminescence intensity of 5TGM1 mice fed with the control diet or serine-free diet at week 6. **h** Platelet counts in PB of 5TGM1 mice fed with the control diet or serine-free diet. **i** Platelet counts in PB of 5TGM1 mice fed with the control diet or control diet (ctrl diet) plus TPO, high-serine diet or high-serine diet plus TPO at 7 weeks. **j** Live imaging of the tumor-associated luminescence intensity of 5TGM1 mice fed with the control diet or control diet plus TPO, high-serine diet or high-serine diet plus TPO at week 7. Results represent means \pm SD; * $p < 0.05$, ** $p < 0.01$, *** $p < 0.001$, ns: not significant ($p > 0.05$). Unpaired two-sided *t*-test were used in **a**, **f**, **h**; One-way ANOVA followed by multiple comparisons were used for **c**, **d**, **i**.

Comment 3: Regarding fig 5 c-e, did authors also try to add methionine to MK with SLC38A1-sh? This would indicate that methionine triggers the same effects as serine.

Response to comment 3: Thanks for your constructive suggestions. We examined megakaryopoiesis and thrombopoiesis by adding methionine (1mM) to $\text{CD}34^+$ cells infected with Scramble or SLC38A1-shRNA virus undergoing megakaryocytic differentiation (Fig. R2a). We

found that megakaryocytic differentiation and platelet production were enhanced with *SLC38A1* knockdown, but the addition of methionine inhibited megakaryocytic differentiation and platelet production in the presence of *SLC38A1* knockdown (Fig. R2b-d, Fig. R2b becomes Supplementary Fig. 5c, Fig. R2d becomes Supplementary Fig. 5d, new data). These results indicate that methionine triggers the same effects as serine. The results, which have been included in the revised manuscript (Supplementary Fig. 5c, d), showed that methionine inhibits megakaryopoiesis and thrombopoiesis *in vitro* (labeled in red in the revised manuscript).

Figure R2. Methionine inhibits megakaryopoiesis and thrombopoiesis *in vitro*

a Schematic diagram of CD34⁺ cells infected with Scramble or SLC38A1-shRNA virus in the presence vehicle or methionine (Met). **b** Generation of CD41a⁺CD42b⁺ MKs of CD34⁺ cells infected with Scramble or SLC38A1-shRNA virus in the presence vehicle or Met. **c** Phase-contrast images of proplatelet formation at day 12 from CD34⁺ cells infected with Scramble or SLC38A1-shRNA virus in the presence vehicle or Met. **d** Generation of CD41a⁺CD42b⁺ PLPs from CD34⁺ cells infected with Scramble or SLC38A1-shRNA virus in the presence vehicle or Met. Results represent means ± SD; **p* < 0.05, ***p* < 0.01, ****p* < 0.001, ns: not significant (*p* > 0.05). Unpaired two-sided *t*-test were used in **b, d**.

Comment 4: It would be interesting to see whether treating mice with a PHGDH inhibitor would lead to 1) less serine in BM plasma and 2) more MK differentiation and less cytopenia.

Response to comment 4: Thanks for your constructive suggestions. We assessed the effect of a PHGDH inhibitor on 5TGM1 mice. NCT-503 was prepared in a vehicle containing 5% PEG400 and 5% Tween 80 and injected intraperitoneally at 40 mg/kg every day as previously described³. The mice were given NCT-503 from week 4 to the end of the experiment (Fig. R3a). Although we found that there was no great significance between the group of the vehicle and PHGDH inhibitor NCT-503, a trend of an increase in platelets and MKs (Fig. R3b, c) and lower bioluminescence intensity in NCT503-treated mice compared with vehicle-treated mice were observed (Fig. R3d, e). Also,

there was a trend for an NCT503-induced reduction in the serum levels of serine (Fig. R3f). Because the difference may be subtle, we envision that a more comprehensive analysis involving the use of a much larger number of mice would be more appropriate for future studies.

Figure R3. The effect of PHGDH inhibitor on megakaryopoiesis and thrombopoiesis in 5TGM1 mice

a Schematic diagram of 5TGM1 mice treated with the vehicle (n = 4) or NCT503 (n = 4). **b** The proportion of MKs of 5TGM1 mice in group treated with vehicle and NCT503. **c** The proportion of platelets of 5TGM1 mice in group treated with vehicle and NCT503. **d** Live imaging of the tumor-associated luminescence intensity of 5TGM1 mice treated with vehicle or NCT503. **e** The fluorescence intensity of live imaging of 5TGM1 mice in group treated with vehicle and NCT503. **f** The serine level in PB of 5TGM1 mice treated with vehicle or NCT503. Results represent means \pm SD; * p < 0.05, ** p < 0.01, *** p < 0.001, ns: not significant (p > 0.05). Unpaired two-sided t -test were used in **b**, **c**, **e**, **f**.

Comment 5: The findings regarding the use of TPO are somewhat surprising since a recent paper demonstrates the opposite: <https://doi.org/10.1016/j.ajpath.2020.12.016>. In this paper, authors demonstrate that TPO is upregulated during MM progression and is negatively correlated to platelet count. Moreover, this TPO was found to be pro-tumoral. Can the authors comment on this? It is also not clear why TPO treatment would result in less serine in the serum when used during treatment with a high serine diet (fig 7b). What cells take up the serine because of the TPO treatment? Why was there no group with only TPO treatment?

Response to comment 5: Thank you for this valuable information. In this recent paper, the authors found that TPO is elevated in MM patients compared with monoclonal gammopathy of undetermined significance/smoldering MM patients and correlates negatively to platelet count. In

addition, TPO promotes blood vessel formation *in vitro* and *in vivo*. The authors proposed that TPO accelerates MM progression by promoting angiogenesis. Although the level of TPO is correlated with MM progression, its role in which stage is not clarified. In our study, we found that among 1,468 patients, 906 (61.7%) had normal platelet counts, while 446 (30.4%) and 116 patients (7.9%) had low and high platelet counts, respectively. Furthermore, the subgroup of patients with low platelet and high serine exhibits impaired megakaryopoiesis and thrombopoiesis. Therefore, TPO and serine intervention are mainly focused on this subgroup of patients in our study. Moreover, it will be more efficient to relieve MM progression at earlier stage (at 2.5 weeks in Fig. 7a in the revised manuscript) by increasing the number of MKs and platelet counts. It is thus possible that the discrepancy between this earlier study and ours may be due to the differences in the patient subgroups.

For the reason that serine level is reduced in 5TGM1 mice after TPO intervention, we think that the possible explanation is that TPO delays MM progression. It is well known that there are two major sources of serine *in vivo* – from the diet or synthesized by cells. In previous studies, we found that serine is elevated in 5TGM1 mice with the progression of MM, suggesting that serine can be used as a marker for disease progression. On the other hand, the finding that serine is elevated *in vivo* at the late stage of MM suggests a potential important role for endogenous serine synthesis.

We apologize for missing out this group of TPO analysis and appreciate your constructive suggestions. Earlier, we mainly sought to explore the effect of combined TPO and serine. To address the Reviewer's point, we have performed the experiment again and added the group of TPO (Fig. R4a). The serine concentration in the serum was elevated in mice fed with high-serine diet compared with control diet, but was reduced in mice either with control diet or high-serine diet combined with TPO intervention (Fig. R4b). The numbers of MKs and platelets in mice were also restored to higher levels with TPO intervention (Fig. R4c, d). In addition, quantitative imaging analysis revealed enhanced infiltration of tumor cells and tumor burden after mouse feeding with high-serine diet, which was reduced after the administration of TPO (Fig. R4e, f). IgG2b level was also reduced in 5TGM1 mice with TPO intervention (Fig. R4g). These results suggest that TPO administration suppresses MM progression at least partially by facilitating megakaryopoiesis and thrombopoiesis. The results have been included in the revised manuscript (Fig. 7a-g, new data) and are labeled in red.

Figure R4. TPO administration lessens thrombocytopenia and suppresses MM progression

a Schematic diagram of 5TGM1 mice fed with the control diet (ctrl diet, n = 6), control diet plus TPO intervention (n = 6), high-serine diet (n = 6) or high-serine diet plus TPO intervention (1 µg/mouse/d) (n = 6). **b** Serum serine concentrations in 5TGM1 mice fed with the indicated diets with BSA or TPO intervention. **c** Platelet counts in PB of 5TGM1 mice fed with the indicated diets with BSA or TPO intervention. **d** The percentage of MKs in BM cells of 5TGM1 mice fed with the indicated diets with BSA or TPO intervention as revealed with flow cytometry analysis. Unpaired two-sided *t*-test was used in (b, c, d). **e** Live imaging of the tumor-associated luminescence intensity of 5TGM1 mice fed with the control diet, control diet plus TPO, high-serine diet or high-serine diet plus TPO at week 7. **f** Quantification of tumor-associated luminescence intensity in the 5TGM1 mice cohorts shown in panel e (mean ± SD). Unpaired one-sided *t*-test was used in (f). **g** Serum IgG2b levels in 5TGM1 mice fed with the control or high-serine diet with or without TPO. Results represent means ± SD; **p* < 0.05, ***p* < 0.01, ****p* < 0.001, ns: not significant (*p* > 0.05). Data were analyzed by One-way ANOVA followed by multiple comparisons in b, c, d, g.

Minor comment

Comment 6: Typo: p 12 first line, this should not read a reduction in serum levels but rather an increase in serum levels.

Response to comment 6: Thank you for pointing it out. We have corrected this mistake in the revised manuscript.

References:

1. Xia, J., *et al.* Blocking glycine utilization inhibits multiple myeloma progression by disrupting glutathione balance. *Nat Commun* **13**, 4007 (2022).
2. Jeong, J.Y., *et al.* The non-peptide thrombopoietin receptor agonist eltrombopag stimulates megakaryopoiesis in bone marrow cells from patients with relapsed multiple myeloma. *J Hematol Oncol* **8**, 37 (2015).
3. Wei, L., *et al.* Genome-wide CRISPR/Cas9 library screening identified PHGDH as a critical driver for Sorafenib resistance in HCC. *Nat Commun* **10**, 4681 (2019).

Reviewer #2 (Remarks to the Author); expert in megakaryocytes:

This is an interesting study that describes novel mechanisms that play a role in thrombocytopenia in multiple myeloma patients by analyzing blood and bone marrow from a mouse model and from patients. The paper shows that thrombocytopenia in MM patients correlates with a poor prognosis. Excessive serine, which is released from MM cells into the bone marrow microenvironment, is identified as the main inducer of thrombocytopenia through suppression of thrombopoiesis and induction of platelet apoptosis in the peripheral blood. Mechanistically, serine is shown to downregulate the actin-binding protein Supervillin via S-adenosyl-methionine (SAM)-mediated trimethylation of H3K9, leading to impaired thrombopoiesis. Finally, the authors showed that TPO administration or serine-free diet might restore the physiological thrombopoiesis and improve survival in a MM mice model.

The authors present well-controlled experiments, and sound methodology and illustrate their findings extensively. Data is novel and improves the understanding of the pathophysiology of thrombocytopenia in this disease. However, I have a couple of general comments for re-consideration by the authors and also some specific comments on areas that require clarification and/or additions.

Major comment

Comment 1: In the first part of the paper, when assessing clinical and biological variables, platelet counts below $100 \times 10^9/L$ were considered for the “low platelets” group. While in other analyses an upper limit of $120 \times 10^9/L$ in the “low platelets” group was adopted. Why this discrepancy?

Response to comment 1: We appreciate your valuable comments. Generally, thrombocytopenia is defined as platelets less than $100 \times 10^9/L$, while some studies define thrombocytopenia as under

150×10⁹/L. In the first part of the paper, to assess the clinical and biological variables from larger cohort and multiple centers, the standard criteria - below 100×10⁹/L for analysis of clinical and biological variables are implied. While in other analyses, the samples used for un-targeted metabolomics were from a single center, Xiangya Hospital, CSU, China, where patients with thrombocytopenia are defined as platelets less than 120×10⁹/L due to differences in instruments and detection techniques. Considering the sample size for un-targeted metabolomics, we eventually selected 120×10⁹/L as cutoff value in this study. Thanks to the Reviewer’s concerns, we also rechecked and reassessed the clinical and biological variables of 1,468 MM patients from multiple centers, and platelet counts below 120×10⁹/L are now defined as the “low platelets” group. The results are also consistence if we define thrombocytopenia as platelet less than 120×10⁹/L (Fig. 1c,d are replaced by Fig. R5a, b, new data).

Figure R5. Thrombocytopenia links to poor prognosis in MM patients

a Kaplan-Meier analysis of overall survival (OS) in NDMM patients with normal platelet counts (120-300×10⁹/L), low platelet counts (50-120×10⁹/L), and low platelet counts (≤50×10⁹/L) at different ISS stages. **b** Kaplan-Meier analysis of progression-free survival (PFS) in MM patients with normal platelet counts (120-300×10⁹/L), low platelet counts (50-120×10⁹/L), and low platelet counts (≤50×10⁹/L) at different ISS stages. *p* values were calculated using Log-rank test.

Comment 2: Were treated-patients excluded from the cohort?

Response to comment 2: We appreciate your valuable comments. In this study, we mainly focus on the pathogenesis of thrombocytopenia in newly diagnosed MM patients. Therefore, MM patients

received treatment are excluded in our study.

Comment 3: How were MKs quantified in BM aspirates? A H&E image is included in Fig S1F but absolute numbers are presented in Fig 1e-f. Immunostaining of sections of bone marrow from patients would be more appropriate.

Response to comment 3: Thank you very much for your constructive comments and suggestions. We quantified MK numbers, which can be distinguished from other cells in morphology, by staining with Wright-Giemsa, in BM aspirates derived from MM patients. Therefore, retrospective studies of the MKs were quantified by staining with Wright-Giemsa in Fig. 1e-g (the previous manuscript). HD with platelets more than $300 \times 10^9/L$ are excluded in our revised manuscript. Consistent with previous studies, the patients at stage II and III showed significantly lower amounts of MKs (Fig. R6a becomes Supplementary Fig. 1d). Furthermore, fewer MKs were found in the patients with lower count of platelets, and the number of MKs and platelets correlated positively ($n = 114$, $r = 0.3266$, $p = 0.0004$) (Fig. R6b, c become Supplementary Fig. 1e, f). The number of MKs was significantly lower in 5TGM1 mice with low platelet counts than those with normal platelet counts, as revealed by H&E staining (Fig. R6d, e, previous Fig. S1f, g). The results have been included in the revised manuscript (Supplementary Fig. S1i, j).

We agree with the constructive suggestion that immunostaining is a more accurate method for detecting the number of MKs in the bone marrow. In retrospective studies on MM patients with or without thrombocytopenia ($n=44$) from paraffin-embedded tissues, immunohistochemical staining of CD41 (Anti-CD41 antibody, ab134131, Abcam) was applied on the section of bone tissues. We found fewer MKs in the patients with a lower count of platelets (Fig. R6f, g become Fig. 1e, f, new data), and the number of MKs and platelets correlated positively ($n=44$, $r = 0.5141$, $p = 0.0004$) (Fig. R6h becomes Fig. 1g, new data). Thus, these results suggest that megakaryopoiesis is impaired in MM patients.

Figure R6. Thrombocytopenia links to poor prognosis in MM patients

a The number of MKs in BM aspirate smears derived from healthy donors (HD), MGUS (monoclonal gammopathy of undetermined significance) and NDMM patients with different ISS stage (nHD = 16, nMGUS=4, nMM(ISS I) = 14, nMM(ISS II) = 31; nMM(ISS III) = 45) by using Wright-Giemsa staining. **b** The number of MKs in BM aspirate smears derived from HD and MM patients with normal-platelet counts and low-platelet counts (mean \pm SD, nHD = 16, nMM(normal) = 75, nMM(low) = 23). **c** Assessment of the correlation between platelet counts and numbers of MKs in BM (n = 114). **d** Representative images of MKs in the BM from 5TGM1 mice with normal platelet counts or low platelet counts by using Hematoxylin staining. **e** Statistical analysis of the number of MKs per mm² in the BM of 5TGM1 mice with normal platelet counts (n = 4) or low platelet counts (n = 3). **f** Representative images of CD41⁺ MKs in the BM from MM patients with normal platelet counts or low platelet counts by using immunohistochemical staining. **g** Statistical analysis of the number of CD41⁺ MKs per 100 mm² in the BM of MM patients with normal platelet counts (n = 25) or low platelet counts (n = 19). **h** Assessment of the correlation between platelet counts and numbers of CD41⁺ MKs in BM (n = 44). Results represent means \pm SD; * p < 0.05, ** p < 0.01, *** p < 0.001, ns: not significant (p > 0.05). Unpaired two-sided t -test were used in **a**, **b**, **e**, **j**; Two-sided Pearson test were used in **c**, **h**.

Comment 4: How were *in vitro* platelets/PLP quantified? This is a critical point.

Response to comment 4: We assessed *in vitro* platelets/PLPs with flow cytometry using the PLP surface marker of CD41a⁺CD42b⁺ as we previously described¹. The gate strategy is such that we first gated the CD41a⁺CD42b⁺ platelets of FSC^{low}SSC^{low} population from PB platelets (Fig. R7a) and then detected the proportion of CD41a⁺CD42b⁺ platelets of FSC^{low}SSC^{low} population from cultured MKs (Fig. R7b).

Figure R7. The gate strategy of PLP *in vitro*

a The gate strategy of CD41a⁺CD42b⁺ platelet from PB by using flow cytometry. **b** The gate strategy of CD41a⁺CD42b⁺ PLP from culture megakaryocytes by using flow cytometry.

Comment 5: Is serine affecting MK differentiation or specifically platelet release? In the MM mouse model, it seems that also hematopoietic progenitors are affected by disease progression. How specific is this mechanism for megakaryocytes? Does serine affect also MK ploidy?

Response to comment 5: Thanks for your constructive comments. We found that the proportion of MKs and hematopoietic progenitors is reduced at the late stage of the disease in 5TGM1 mouse model (Fig S1). To explore whether serine affects MKs or hematopoietic progenitors, we fed 5TGM1 mice with control diet or serine-free diet. Interestingly, we found that the number of platelets and MKs was elevated in 5TGM1 mice fed with serine-free diet compared with the control diet (Fig. R8a), while the proportion of LSKs, CMPs and MEPs was unaffected (Fig. R8b becomes Supplementary Fig. 3g, new data). To assess whether serine affects MK ploidy, we stained cells from MK differentiation with Hoechst 33422. However, MK ploidy was not affected by serine treatment (Fig. R8c becomes Supplementary Fig. 3b, new data). These results indicate that serine affects MK differentiation and platelet release, but not hematopoietic progenitors. In addition, serine did not affect MK ploidy.

Figure R8. Serine affects MK differentiation and platelet release

a The gate strategy of HSPCs with flow cytometry. **b** The proportion of LSK⁺ cells, CMP, MEP and MKs in BM from 5TGM1 mice fed with control diet and serine-free diet. **c** The proportion of MK ploidy including 2n, 4n and 8n in CD41a⁺ CD42b⁺ cells treated with vehicle or different concentration of serine. Results represent means ± SD; * $p < 0.05$, ns: not significant ($p > 0.05$). Data were analyzed by unpaired two-sided t -test in **b**; One-way ANOVA followed by multiple comparisons in **c**.

Comment 6: Serine-induced apoptosis in platelets. Is there a similar effect on cultured megakaryocytes?

Response to comment 6: Thanks for your constructive suggestion. We evaluated the apoptosis of MKs in the presence of various concentrations of serine with flow cytometry (Fig. R9a). Consistence with the increasing platelet apoptosis caused by serine, the apoptosis of MKs was also elevated in the presence of excessive serine (Fig. R9b becomes Supplementary Fig. 3c, new data).

Figure R9. The apoptosis of MKs was enhanced in the presence of serine

a The representative images of apoptosis in CD41a⁺CD42b⁺ cells treated with vehicle or different concentration of serine as detected by flow cytometry. **b** The proportion of apoptosis in CD41a⁺CD42b⁺ cells treated with vehicle or different concentration of serine as detected by flow cytometry. Results represent means \pm SD; * $p < 0.05$, ns: not significant ($p > 0.05$). Data were analyzed by One-way ANOVA followed by multiple comparisons in **b**.

Comment 7: Figure 3e-f: Luminescence intensity and relative quantification are quite different. In the serine-deficient diet condition, it seems that all the mice display a very low signal in the histogram, while 50% of mice have substantial luminescence in the photographs. Why is this the case?

Response to comment 7: Thanks for your constructive comments. we apologize for the inaccurate presentation of Fig.3e & Fig.3f. Although it seems that 50% of the mice fed with serine-free diet have substantial luminescence in the photographs, the statistics of overall luminescence intensity in mice fed with serine-free diet was lower than mice fed with the control diet. To illustrate the differences more accurately between the two groups, we adjusted the fluorescence intensity ranging from 1.74e5-9.28e7 to 7.16e5-3.60e7 at week 6 (Fig. R10a, Fig. 3e in the revised manuscript). Moreover, we have also improved the display of Fig. 3f (Fig. R10b, Fig. 3f in the revised manuscript). The new results have been included in the revised manuscript (Fig. 3e, f).

Figure R10. The progression is delayed in 5TGM1 mice fed with serine-free diet

a Tumor-associated luminescence intensity in live 5TGM1 mice **b** The luminescence intensities of 5TGM1 mice in group fed with control diet and group fed with serine-free diet at week 2, 4 and 6. Results represent means \pm SD; * p < 0.05, ns: not significant (p > 0.05). Unpaired two-sided t -test were used in **b**.

Comment 8: Figure 3B: Was proplatelet formation analyzed on day 15 (figure label) or day 12 (figure legend)?

Response to comment 8: Thanks for your comment. We apologize for this mistake. Proplatelet formation was analyzed at day 15. We corrected it in our revised manuscript.

Comment 9: Is SLC38A mainly expressed on MKs? What about hematopoietic progenitors? Is the actin-binding protein Supervillin mainly expressed by MKs? In the early or late phases of thrombopoiesis?

Response to comment 9: Thanks for your insightful questions. To explore the expression of SLC38A1 and Supervillin (SVIL) during MK differentiation, we examined mRNA levels of *SLC38A1* and *SVIL* in the CD34⁺ cells undergoing MK differentiation at day 6, 9 and 12. We also utilized two public GEP databases (accession number: GSE24870 and GSE124310) including the expression of *SLC38A1* and *SVIL* in HSPCs as well as in other cells in BMME at single-cell level from HD and MM^{2,3}. MKs was generated at day 9 during MK differentiation and became mature at day 12 as revealed by staining with Wright-Giemsa (Fig. R11a). Therefore, the cells from day 9 and day 12 could be used as various stages of MKs. The results showed that *SLC38A1* is mainly expressed in progenitors and are gradually downregulated at day 9 and day 12 during megakaryocytic differentiation (Fig. R11b, c), while *SVIL* expression increased with megakaryocyte differentiation (Fig. R11d, e) at both the mRNA and protein levels. Moreover, the expression of

SLC38A1 and *SVIL* in hematopoietic stem cells of MM patients is not significantly altered compared with HDs (GSE24870) (Fig. R11f & g). The expression of *SLC38A1* is elevated in MKs derived from MM compared with HD (Fig. R11h), while *SVIL* is not detected in this database (Fig. R11i). Together, these results suggest that *SLC38A1* is mainly expressed in the progenitor cells, but the expression of *SLC38A1* is up-regulated in megakaryocytes in MM, which may be linked to the increased serine level in the microenvironment. *SVIL* is mainly expressed in the megakaryocytes.

Figure R11. The expression of *SLC38A1* and *SVIL* in HSPCs and MKs from HD and MM

a The morphology of cells undergoing megakaryocytic differentiation at day 6, 9, 12 by using staining with Wright-Giemsa. **b** The mRNA level of *SLC38A1* in cells undergoing megakaryocytic differentiation at day 6, 9 and 12 by using RT-qPCR. **c** The protein level of *SLC38A1* in cells undergoing megakaryocytic differentiation at day 6, 9 and 12 by using RT-qPCR and western blotting. **d** The mRNA level of *SVIL* in cells undergoing megakaryocytic differentiation at day 6, 9 and 12 by using RT-qPCR. **e** The protein level of *SVIL* in cells undergoing megakaryocytic differentiation at day 6, 9 and 12 by using western blotting. **f** The mRNA expression of *SLC38A1* in HSC、CMP and MEP derived from HD and MM patients. **g** The mRNA expression of *SVIL* in HSC、CMP and MEP derived from HD and MM patients. **h** The mRNA expression of *SLC38A1* in CD41⁺ cells derived from HD and MM patients. **i** The mRNA expression of *SVIL* in CD41⁺ cells derived from HD and MM patients. Results represent means \pm SD; * $p < 0.05$, ** $p < 0.01$, *** $p < 0.001$, ns: not significant ($p > 0.05$). Unpaired two-sided *t*-test were used in **b, d**.

Comment 10: The origin of serine from MM cells is another critical aspect. Non-hematopoietic cellular components of the BM were not considered. Are levels of PHGDH and PSPH significantly

different in MM cells among patients with low/normal platelet counts?

Response to comment 10: Thanks for your insightful questions. There are two cell types in BM including hematopoietic cells, which are differentiated from HSCs, and non-hematopoietic cells, among which the major cell type is BMSCs (Fig. R12a). To explore the origin of serine, we first isolated the hematopoietic cellular components including progenitors (Lin⁻c-Kit⁺ cells, LK), lymphocytes (B cells, B; plasma cells, P and T cells, T), myeloid cells (Erythrocytes, Ery) as well as the non-hematopoietic cells BMSCs (Fig. R12b becomes Fig. 6a, new data) using their respective surface markers. Then, we examined the expression of mRNA and protein level of *Phgdh* in LK, B, plasma cells, T, erythrocytes and BMSCs by using RT-qPCR and immunofluorescence. As expected, both the mRNA and protein levels of *Phgdh* are higher in plasma cells (MM cells) than other cells (Fig. R12c becomes Fig. 6b, Fig. R12d becomes Fig. 6c, Fig. R12e becomes supplementary Fig.7a, new data). These results indicate that serine synthesis is extremely active in plasma cells (MM cells). We further employed immunofluorescence to assess the expression of PHGDH in CD138⁺ MM cells from the bone marrow of MM patients with normal platelet count (n=6) and low platelet count (n=4). As expected, the protein level of PHGDH is higher in CD138⁺ MM cells derived from MM patients with low platelet count than those with normal platelet count (p=0.0435, Fig. R12f, Fig. R12f becomes Supplementary Fig. 7b, new data). Therefore, PHGDH is significantly elevated in MM cells from patients with low platelet counts compared with those with normal platelet counts.

Figure R12. PHGDH is highly expressed in plasma cells from MM patients with thrombocytopenia

a Hematopoietic cellular components and non-hematopoietic cellular components in BM. **b** The gate strategy of LK, B, P, T, Ery and BMSCs with flow cytometry. **c** The mRNA expression of *Phgdh* and *Psph* in LK cells, B, plasma cells (P), T, erythrocytes (Ery) and BMSCs (gating in CD34⁺CD45⁻ cells) derived from 5TGM1 mice, BMSCs was used as control group. **d** The fluorescence image of Phgdh in LK cells, B, P, T, Ery and BMSCs derived from 5TGM1 mice, BMSCs was used as control group. **e** The fluorescence intensities of Phgdh in LK cells, B, P, T, Ery and BMSCs derived from 5TGM1 mice. **f** The fluorescence intensities of PHGDH in CD138⁺ cells derived from MM patients with normal platelet counts and low platelet counts. * $p < 0.05$, ** $p < 0.01$, *** $p < 0.001$, ns: not significant ($p > 0.05$). Unpaired two-sided *t*-test were used in **f**; One-way ANOVA followed by multiple comparisons were used for **c**, **e**.

Comment 11: The last experimental point in the MM mice model is at week 6 throughout the manuscript. However, the effects of TPO administration were recorded until week 8 (Fig 7C). Can the authors explain the experiment design?

Response to comment 11: We appreciate your valuable comments. To prepare 5TGM1 mice models, we pretreated C57BL/KaLwRijHsd mice with control diet or serine-free diet for 1 week.

1×10^6 luciferase-labeled 5TGM1 cells were then injected into C57BL/KaLwRijHsd mice via the tail vein. After 2.5 weeks, mice in control group or serine-free group were randomly divided into two subgroups (six mice in each group), and then treated with 1% BSA or TPO (1 $\mu\text{g}/\text{mouse}/\text{d}$) subcutaneously for 4 consecutive days. The clinical end point was achieved when mice exhibited signs of hindlimb weakness or when the overall fluorescence intensity was around 1×10^9 . Generally, TPO could sustain its effects on megakaryopoiesis and thrombopoiesis for 2 to 3 weeks. To better explore the effect of increased MKs and platelets on MM progression, we extended the time to week 8 for observations.

Figure R13. Schematic diagram of work flow in 5TGM1 mice model

References:

1. Wang, H., et al. MEIS1 Regulates Hemogenic Endothelial Generation, Megakaryopoiesis, and Thrombopoiesis in Human Pluripotent Stem Cells by Targeting TAL1 and FLI1. *Stem Cell Reports* **10**, 447-460 (2018).
2. Bruns, I., et al. Multiple myeloma-related deregulation of bone marrow-derived CD34(+) hematopoietic stem and progenitor cells. *Blood* **120**, 2620-2630 (2012).
3. Zavidij, O., et al. Single-cell RNA sequencing reveals compromised immune microenvironment in precursor stages of multiple myeloma. *Nat Cancer* **1**, 493-506 (2020).

Reviewer #3 (Remarks to the Author); expert in metabolism:

The decrease in platelet counts is associated with poor prognosis in patients with multiple myeloma (MM). Kuang, Xia, and colleagues report that increased serine levels prevent megakaryopoiesis and thrombopoiesis and eventually cause thrombocytopenia. The influence of high serine concentrations on thrombocytopenia is mediated via the repression of megakaryocyte (MK) differentiation in the bone marrow and the induction of apoptosis of platelets in the peripheral blood (PB). The authors claim that the serine molecules are transported into MKs via SLC38A1 and decrease the actin-binding protein Supervillin (SVIL) via SAM-mediated tri-methylation of H3K9, resulting in the inhibition of megakaryopoiesis and thrombopoiesis.

This is an exciting study, but the current data are not sufficient to support the claim made by the authors.

Comment 1: This study is unfortunately unclear. On the one hand, the authors claim that serine enters the cells through the upregulation of SLC38A1, which is presumed to be important for regulating MK differentiation and thrombopoiesis. On the other hand, PHGDH is upregulated to promote serine synthesis and supports the repression of MK differentiation and thrombopoiesis. It is difficult to understand which pathway is essential here. Is it serine uptake or serine synthesis through PHGDH?

Response to comment 1: Thanks for your insightful questions. There are two cell types in BM including hematopoietic cells, which are differentiated from HSCs, and non-hematopoietic cellular cells, among which the major cell type is BMSCs (Fig. R14a). To explore the cell origin of serine, we first isolated hematopoietic cellular components including progenitors (lin⁻c-Kit⁺ cells, LK), lymphocytes (B cells, B; plasma cells, P and T cells, T), myeloid cells (Erythrocytes, Ery) as well as non-hematopoietic cells BMSCs according to their surface markers (Fig. R14b becomes Fig. 6a, new data) and then assessed the expression of mRNA and protein level of *Phgdh* in LK, B, plasma cells, T, erythrocytes and BMSCs by using RT-qPCR and immunofluorescence. These results showed that both the mRNA (Fig. R14c becomes Fig. 6b, new data) and protein levels of *Phgdh* are higher in plasma cells (MM cells) than in other cells (Fig. R14d becomes Fig. 6c and Supplementary Fig. 7a, new data). Furthermore, our previous study demonstrated that the serine synthesis pathway¹ and glycine cleavage system² in MM cells are extreme active and contribute to MM cell proliferation and drug resistance. We next directly assessed whether serine was indeed secreted from MM cells. We measured ¹³C-labeled serine by exploring ¹³C₂-glycine metabolic flux of ARP1 cells and found that ¹³C-labeled serine was gradually elevated in the CM of ARP1 cells (Fig. R14e, f, Fig. R14f becomes Fig. 6g). We then evaluated the effects of manipulating PHGDH expression on serine production, megakaryopoiesis and thrombopoiesis by generating APR1 myeloma cells with PHGDH ectopic expression or knockdown. As expected, CM collected from PHGDH-overexpressing APR1 cells inhibited megakaryocytic differentiation and PLP generation (p<0.05) (Fig. R14g becomes Fig. 6j in manuscript). In contrast, knockdown of PHGDH decreased

the level of serine in the CM and facilitated megakaryocytic differentiation and PLP derivation (Fig. R14h becomes Fig6m in manuscript). Therefore, these results suggest that serine accumulated in the BM microenvironment is released from MM cells with high PHGDH and that serine uptake is mediated by SLC38A1 in MK cells to inhibit megakaryocyte differentiation and platelet production (Fig. R14i). The revised results have been included in the revised manuscript (Fig. 6a-c and Supplementary Fig. 7a).

Figure R14. MM cells derived serine was uptake by megakaryocytes

a Hematopoietic cellular components and non-hematopoietic cellular components in BM. **b** The gate strategy of LK,

B, P, T, Ery and BMSCs with flow cytometry. **c** The mRNA expression of *Phgdh* and *Psph* in in LK cells, B, Plasma cells (P), T, Erythrocytes (Ery) and BMSCs (gating in CD34⁺CD45⁻ cells) derived from 5TGM1 mice. **d** The fluorescence image (left) and fluorescence intensities (right) of Phgdh in LK cells, B, P, T, Ery and BMSCs derived from 5TGM1 mice. **e** The metabolic flux of conditioned medium of labeled serine in ARP1 cells incubated with ¹³C₂-glycine. **f** ¹³C-labeled serine derived from ¹³C₂-glycine in conditioned medium of ARP1 cells. **g** Generation of CD41a⁺CD42b⁺ PLPs at day 12 from CD34⁺ cells treated with CM derived from ARP1-EV and ARP1-PHGDH cells. **h** Generation of CD41a⁺CD42b⁺ PLPs at day 12 from CD34⁺ cells treated with ARP1-PHGDH-sh1 CM and ARP1-PHGDH-sh2 CM. **i** The origin of serine and the utilization of serine by MKs. Serine is synthesized by PHGDH and secreted by MM cells, then the extracellular serine was uptake by MKs. Results represent means ± SD; **p* < 0.05, ***p* < 0.01, ****p* < 0.001, ns: not significant (*p* > 0.05). Unpaired two-sided *t*-test were used in **g**, **h**; One-way ANOVA followed by multiple comparisons were used for **c**, **d**.

Comment 2: SLC38A1 is not the canonical transporter of serine. So why is SLC1A4/5 not involved?

Do SLC1A4/5 knockdowns increase MK differentiation and H3K9me?

Response to comment 2: Thanks for your insightful question. There are four transporters of serine including SLC1A4/5 and SLC38A1/2. To explore which transporter(s) is involved in MK differentiation, we performed RNA-sequencing analysis of CD34⁺ cells treated with vehicle or serine and then validated the mRNA and protein expression level of these four transporters by using RT-qPCR and western blotting, respectively. We found that only the *SLC38A1* is highly elevated at both mRNA and protein levels in MKs treated with serine (Fig. R15a-c, Fig. R15a becomes Fig. 4a, Fig. R15b becomes Fig. 4c, Fig. R15c becomes Supplementary Fig. 4a, new data). Therefore, we speculate that that SLC38A1 may be the main transporter that mediates serine entry into the MKs.

Figure R15. The mRNA and protein levels of *SLC38A1* is upregulated in MKs treated with serine

a The expression profile of serine metabolism-related genes in MKs cultured with vehicle or excessive serine. **b** Assessment of mRNA serine transporters *SLC1A4*, *SLC1A5*, *SLC38A1* and *SLC38A2* in differentiated cells at day 12 treated with vehicle or excessive serine by using RT-qPCR. **c** Western blot analysis of serine transporters SLC1A4,

SLC1A5 and SLC38A1 in differentiated cells at day 12 treated with vehicle or excessive serine. Results represent means \pm SD; * $p < 0.05$, ns: not significant ($p > 0.05$). Unpaired two-sided t -test were used in **b**.

Comment 3: The authors did not demonstrate that serine, through feeding SAM synthesis, leads to H3K9me3. This is an important part of this project. Once serine enters the cells, it can be used by the serine hydroxymethyltransferase (SHMT1/2) to support 1C metabolism and nucleotide synthesis. For example, is H3K9me3 demethylated in response to SHMT inhibition? Does SHMT1/2 inhibition lead to the increase of megakaryopoiesis and thrombopoiesis and the inhibition of MM progression?

Response to comment 3: Thanks for your constructive comments. The conversion of serine into one-carbon units requires serine hydroxymethyltransferase (SHMT1/2). According to the studies from other groups, SHMT2 is required for the conversion of serine into glycine and a tetrahydrofolate-bound one-carbon unit (Fig. R16a, Fig. R16a becomes Supplementary Fig. 5e)^{3,4}. To investigate whether excessive serine inhibits megakaryopoiesis and thrombopoiesis through one-carbon metabolism, we infected CD34⁺ cells with small hairpin RNAs (shRNAs) that target *SHMT2*. The results show that the protein levels of SHMT2 and H3K9me3 were both reduced in cells transfected with SHMT2-shRNA (Fig. R16b, Fig. R16b becomes Supplementary Fig. 6c, new data). In addition, knockdown of *SHMT2* promotes megakaryopoiesis and thrombopoiesis (Fig. R16c-e) (Fig. R16c, e become Fig. 5e, f new data). Our data provide evidence that one-carbon metabolism is required for the inhibitory effects of serine on megakaryopoiesis and thrombopoiesis.

Figure R16. Knockdown of *SHMT2* promotes megakaryopoiesis and thrombopoiesis

a Overview of one-carbon metabolism around the folate cycle and methionine cycle. **b** Western blot analysis of *SHMT2* and H3K9me3 in CD34⁺ cells infected with Scramble or *SHMT2*-shRNA virus. **c** Generation of CD41a⁺CD42b⁺ MKs of CD34⁺ cells infected with Scramble or *SHMT2*-shRNA virus in the presence vehicle or serine. **d** Phase-contrast images of proplatelet formation at day 12 from CD34⁺ cells infected with Scramble or *SHMT2*-shRNA virus in the presence vehicle or serine. **e** Generation of CD41a⁺CD42b⁺ PLPs from CD34⁺ cells infected with Scramble or *SHMT2*-shRNA virus in the presence vehicle or serine. $n = 3$ independent experiments; Results represent means \pm SD; * $p < 0.05$, ** $p < 0.01$, *** $p < 0.001$, ns: not significant ($p > 0.05$). Unpaired two-sided t -test was applied in **c** and **e**.

Comment 4: It is unlikely that SAM synthesis, downstream of serine catabolism, can solely explain the regulation of H3K9me3. Other histone and RNA methylation marks should also be affected. Other histone marks such as H3K36me are dynamically methylated in response to SAM abundance. Therefore, the authors should assess H3K36me and other histone marks in response to serine availability.

Response to comment 4: Thanks for your constructive suggestions. SAM is the methyl-donor that regulates the methylation of histone. In previous studies, we found that H3K9me3 is upregulated upon serine stimulation. To further assess whether other histone marks are affected by serine in cultured MKs, we examined the protein level of H3K36me3. As expected, H3K36me3 is also elevated in differentiated cells treated with serine (Fig. R17 becomes Fig. 5i, new data). Our data indicate that both H3K9me3 and H3K36me3 are elevated in response to serine availability.

Figure R17. The protein level of H3K36me3 is upregulated in MKs treated with serine

Comment 5: missing

Comment 6: If the effects are SAM-dependent, then methionine starvation or direct MAT2A inhibition should recapitulate the inhibition of serine utilization.

Response to comment 6: Thanks for your insightful comment. SAM is synthesized by methionine and ATP catalyzed by methionine adenosyl-transferase (MAT2A) (Fig. R18a becomes Supplementary Fig. 5e). To test whether the effects are SAM-dependent, we first infected CD34⁺ cells with small hairpin RNAs (shRNAs) that targeted *MAT2A* (Fig. R18b becomes Supplementary Fig. 5f). The results show that knockdown of *MAT2A* promotes megakaryopoiesis and thrombopoiesis (Fig. R18c, d) (Fig. R18 c, d become Supplementary Fig. 5g, h, new data). We then added 3-Deazaa-denosine (3DZA), an inhibitor that suppresses S-adenosylhomocysteine hydrolase and SAM-dependent methylation reactions (SELLECK, S0787), to CD34⁺ cells undergoing megakaryocytic differentiation. We found that megakaryopoiesis and thrombopoiesis of CD34⁺ cells were impaired by serine, while supplementation with 3DZA restored megakaryopoiesis and thrombopoiesis (Fig. R18 e-g, Fig. R18e becomes Fig. 5g, Fig. R18g becomes Fig. 5h, new data). These data provide evidence that SAM generation is required for the inhibition of megakaryopoiesis and thrombopoiesis by serine.

Figure R18. MAT2A inhibition promotes megakaryopoiesis and thrombopoiesis

a Overview of one-carbon metabolism around the folate cycle and methionine cycle. **b** Assessment of mRNA level of *MAT2A* in differentiated cells infected with Scramble or *MAT2A*-shRNA virus at day 12 by using RT-qPCR. **c** Generation of CD41a⁺CD42b⁺ MKs of CD34⁺ cells infected with Scramble or *MAT2A*-shRNA virus in the presence vehicle or serine. **d** Generation of CD41a⁺CD42b⁺ PLPs from CD34⁺ cells infected with Scramble or SHMT2-shRNA virus in the presence vehicle or serine. **e** Generation of CD41a⁺CD42b⁺ MKs of CD34⁺ cells treated with DMSO or 3-DZA in the presence vehicle or serine. **f** Phase-contrast images of proplatelet formation at day 12 from CD34⁺ cells treated with DMSO or 3-DZA in the presence vehicle or serine. **g** Generation of CD41a⁺CD42b⁺ PLPs from CD34⁺ cells treated with DMSO or 3-DZA in the presence vehicle or serine. *n* = 3 independent experiments; Results represent means ± SD; **p* < 0.05, ***p* < 0.01, ****p* < 0.001, ns: not significant (*p* > 0.05). Unpaired two-sided *t*-test was applied in **b**, **c**, **d**, **e**, **g**.

Comment 7: The concentrations of serine used in cell-cultured are extreme (8 mM). Still, the authors do not show SAM levels in response to serine buildup. Therefore, the authors should trace ¹³C₃-serine into newly synthesized SAM.

Response to comment 7: Thanks for your constructive suggestions. Generally, the concentration of serine used in cell culture ranges from 0.1 to 1 mM when serine is deprived from the control medium. However, the concentration of serine is around 0.5 mM in our control medium. Therefore, in order to mimic the high level of serine in BMME in MM patients, we added additional serine into the culture medium. We also found, based on existing literature, that serine concentrations documented in earlier studies range from 1 mM to 20 mM⁵⁻⁷.

To further explore which pathway serine is linked to, we added $^{13}\text{C}_3$ serine tracer to differentiated cells at day 6, 9, 12 for 2 hours. As shown in Fig. 5a and supplementary Fig. 5b in our revised manuscript, we obtained ^{13}C -labeled serine, glycine, GSH, Met, SAM, SAH and purine metabolites AMP, GMP, IMP (Fig. R19a becomes supplementary Fig. 5b). Of note, we observed the accumulation of intracellular ^{13}C -labeled serine and m+2 glycine, as well as the incorporation of ^{13}C -serine-derived carbons into m+1 to 4 GMP, AMP, SAM, and SAH (Fig. R19b becomes Fig. 5a), indicating serine entry into one-carbon metabolism in cells undergoing megakaryocytic differentiation. Thus, these data suggest that exogenous serine supports the methionine cycle and SAM generation in cells undergoing megakaryocytic differentiation.

Figure R19. Exogenous Serine Supports the Methionine Cycle and SAM Generation in MKs

a The fractions of serine, glycine, glutathione, methionine, SAM, SAH, homocysteine (hCys), uridine monophosphate (UMP), guanosine 5-monophosphate (GMP), adenosine 5-monophosphate (AMP) and inosine-5'-monophosphate (IMP) containing one (m+1), two (m+2), three or more (m>3) or zero (m+0) ^{13}C in $\text{CD}34^+$ cells undergoing MK differentiation at day 6, 9 and 12 exposed to $^{13}\text{C}_3$ serine for 2 hr. **b** The fractions of methionine (m+0,

m+1, m+2, m+3), SAM (m+0, m+1, m+2, m+3, m+4), SAH (m+0, m+1, m+2, m+3, m+4), hCys (m+0) in CD34⁺ cells undergoing MK differentiation at day 6, 9 and 12.

References:

1. Wu, X., *et al.* Phosphoglycerate dehydrogenase promotes proliferation and bortezomib resistance through increasing reduced glutathione synthesis in multiple myeloma. *Br J Haematol* **190**, 52-66 (2020).
2. Xia, J., *et al.* Blocking glycine utilization inhibits multiple myeloma progression by disrupting glutathione balance. *Nat Commun* **13**, 4007 (2022).
3. Yu, W., *et al.* One-Carbon Metabolism Supports S-Adenosylmethionine and Histone Methylation to Drive Inflammatory Macrophages. *Mol Cell* **75**, 1147-1160 e1145 (2019).
4. Zeng, Y., *et al.* Roles of Mitochondrial Serine Hydroxymethyltransferase 2 (SHMT2) in Human Carcinogenesis. *J Cancer* **12**, 5888-5894 (2021).
5. Kim, K.Y., *et al.* l-Serine protects mouse hippocampal neuronal HT22 cells against oxidative stress-mediated mitochondrial damage and apoptotic cell death. *Free Radic Biol Med* **141**, 447-460 (2019).
6. Hwang, S., *et al.* Serine-Dependent Sphingolipid Synthesis Is a Metabolic Liability of Aneuploid Cells. *Cell Rep* **21**, 3807-3818 (2017).
7. Dunlop, R.A. & Carney, J.M. Mechanisms of L-Serine-Mediated Neuroprotection Include Selective Activation of Lysosomal Cathepsins B and L. *Neurotox Res* **39**, 17-26 (2021).

Reviewer #4 (Remarks to the Author); expert in epigenetics and metabolism:

The work by Zhou and colleagues identifies a molecular mechanism driven by excessive levels of serum serine from multiple myeloma (MM) cells in the bone marrow (BM) niche that leads to thrombocytopenia and suppression of megakaryocyte (MK) differentiation via repression of Supervillin (SVIL) expression. This impacts MM progression and survival *in vivo*, and the authors further demonstrate that thrombocytopenia is linked to poor prognosis in MM patients and that administration of TPO enhances megakaryopoiesis and thrombopoiesis.

Overall, this was a very thorough study with extensive experimentation and a well-constructed manuscript. This work provides a unique and important molecular mechanism in MM that contributes to co-morbidities that the authors have demonstrated impact overall survival. I have several points that need to be addressed by the authors.

Major points:

Comment 1: Why are ISS I and ISS II groups in Figures 1c-d combined? Would it not be more appropriate to split them into separate figures?

Response to comment 1: Thanks for your insightful questions. The classifications of groups are as follows: ISS I (($n < 50$ (3), n 50-100 (21), n 100-300 (104)); and ISS II (($n < 50$ (11), $50 < n < 100$ (35), n 100-300 (191)). Considering the small population of ISS I and ISS II groups due to the delay of seeking doctor's treatment and having only 3 patients with severe thrombocytopenia (platelets $< 50 \times 10^9/L$) at the ISS stage I, which was insufficient for survival analysis, patients with stage I and II were combined for subsequent analysis. Because of the sample size for un-targeted metabolomics, we eventually selected $120 \times 10^9/L$ as the cutoff value in this study. We also rechecked and reassessed the clinical and biological variables of 1,468 MM patients from multiple centers and decided to use platelet counts below $120 \times 10^9/L$ for the "low platelets" group.

Figure R20. Low platelet counts is correlated with poor prognosis of MM patients

a Kaplan-Meier analysis of overall survival (OS) in NDMM patients with normal platelet counts ($120-300 \times 10^9/L$), low platelet counts ($50-120 \times 10^9/L$), and low platelet counts ($\leq 50 \times 10^9/L$) at different ISS stages. **b** Kaplan-Meier analysis of progression-free survival (PFS) in MM patients with normal platelet counts ($120-300 \times 10^9/L$), low platelet counts ($50-120 \times 10^9/L$), and low platelet counts ($\leq 50 \times 10^9/L$) at different ISS stages. * $p < 0.05$, ** $p < 0.01$, *** $p < 0.001$, ns: not significant ($p > 0.05$). Log-rank test were used in **a**, **b**.

Comment 2: The authors show that serum from MM patients leads to higher platelet apoptosis versus serum from healthy donors (Figure 3j). However, serum serine seems to vary greatly in MM patients in Fig 3k. It would therefore strengthen the authors argument if they can show differences in platelet apoptosis using MM patient serums with widely different serine concentrations.

Response to comment 2: Thanks for your constructive comments. We treated platelets with serum of the patients (n=28) with different serine levels and assessed apoptosis of platelets. As expected, the apoptotic rate of platelets was elevated when they were treated with the serum of MM patients with high-level serine (n=15) compared with low-level serine (n=13) in serum (p=0.0454, Fig. R21 becomes Supplementary Fig. 3h, new data). Therefore, we conclude that the high serine level in serum induces apoptosis of platelets in MM patients.

Figure R21. The apoptosis of platelets is elevated treated with serum from MM patients with high serine

Comment 3: Regarding the RNA-seq data in MK cells, how many replicates were performed for serine vs vehicle treatment groups? What was the fold change differences and p-values for SLC38A1, MTHFD1L, GCSH, PHGDH and the 15 genes that were also identified using ATAC-seq data? Did SLC38A1 protein levels increase as well? The authors also denote a p-value cutoff of 0.05, but do not denote if multiple testing correction was performed on the differentially expressed genes from RNA-seq. If no correction was performed, the authors should try to verify differential expression of relevant genes they follow-up on or describe via RT-qPCR considering the higher false positive rate.

Response to comment 3: Thanks for your insightful questions. RNA-seq was performed in 2 replicates. Therefore, we apologize for the inaccurate denotation. RNA-seq was performed on BGISEQ-500 (Beijing Genomics Institute at Wuhan, China). All sequencing reads were aligned with the reference genome (GRch38) using HISAT2, with default options in the StringTie RNA-seq workflow. After the removal of improperly aligned reads, FPKM of each gene was generated by StringTie with a provided Python script. The fold change of FPKM in the serine group and the vehicle group higher than 1.2 is considered as the cutoff for differentially expressed genes. We did not perform multiple testing correction, but performed RT-qPCR to validate the expression of

potential target genes selected from RNA-seq and ATAC-seq. We found that the levels of *SLC38A1*, *MTHFD1*, *GCSH*, *PHGDH* are upregulated upon serine stimulation, while *SHANK3*, *MELK*, *STRN3*, *FAM193A*, *FRYL*, *SVIL* and *MAX* are downregulated (Fig. R22a becomes Supplementary Fig. 6e). In addition, we also found the protein level of SLC38A1 is upregulated in cells treated with serine (Fig. R22b becomes Supplementary Fig. 4a, new data).

Figure R22. SLC38A1 is elevated in cells treated with serine

a Assessment of mRNA levels of targeted genes in differentiated cells treated with vehicle or excessive serine at day 12 by using RT-qPCR. The mRNA expression of *IL17RE* is unmeasurable in differentiated cells. **b** Western blot analysis of SLC38A1 in differentiated cells at day 12 treated with vehicle or serine. $n = 3$ independent experiments; Results represent means \pm SD; * $p < 0.05$, ** $p < 0.01$, *** $p < 0.001$, ns: not significant ($p > 0.05$). Unpaired two-sided t -test was applied in **a**.

Comment 4: For the ATAC-seq data (Figure 5i), how many replicates were performed for each group? How was the ATAC-seq integrated with RNA-seq? What constituted a positive result? How were ATAC-seq peaks identified and how were differentially accessible peaks subsequently identified? How were ATAC-seq peaks associated with genes? Did the authors solely focus on promoters? Is there any correlation between ATAC-seq signal alterations and gene expression changes at promoters? The authors should provide more detail of their data integration and provide some evidence for its utility in identifying genes of interest. Were peaks and differentially accessible peaks identified at the *SVIL*, *FRYL* and *SHANK3* promoters? Was the p-value cutoff corrected for multiple testing? A multiple testing correction is appropriate given the number of loci that are likely examined.

Response to comment 4: Thanks for your insightful questions. For the ATAC-seq data, we performed ATAC-seq with one sample for each group. Therefore, we apologize for the inaccurate denotation. Peaks were called ($p < 0.05$) with macs2 software. All reads were extended to the 3' direction with a length of 200bp, and the genome was subjected to sliding window, while the

dynamic λ of the window was calculated. The formula of λ is: $\lambda_{\text{local}} = \lambda_{\text{BG}}$ (λ_{BG} refers to the number of reads on the background area). The p value of this window was calculated and FDR correction on the p value of each window was performed. The default corrected p value (that is, q value) was less than or equal to 0.05, and the region was considered as the peak region. Next, read counts, RPM value and FoldEnrich value of the merged peaks were calculated. The $\log_2\text{FC}$ value was then calculated by using the FoldEnrich value of the peaks of the two groups, $\log_2\text{FC}$ more than 1 was considered as the differentially accessible peak. Differentially accessible peaks were then ascertained using the chip-seeker software. The distance between the peak and the TSS (Transcription start site) of the gene was calculated, and the gene with the nearest TSS distance was considered as the peak-related gene. Finally, we integrated these peak-related gene from ATAC-seq with differentially expressed genes from RNA-seq¹⁻³. We have included the detailed description of ATAC-seq in the “Materials and Methods” section.

Integration of data from ATAC-seq and RNA-seq led us to discover 15 potential targets. We focused on promoters because the level of histone methyltransferases was altered in RNA-seq, indicating that these regions were specifically enriched in serine-treated samples. The peaks of *SVIL*, *FRYL* and *SHANK3* promoters were identified as differentially accessible peaks (Fig. R23 becomes Fig. 5k), because their $\log_2\text{FC}$ values were -1.1240113, -1.1854118, and -1.4015453, which were more than 1 (supplementary S6). Despite the lack of replicates for ATAC-seq, the interaction of specific H3K9me3 regions was confirmed by ChIP-qPCR analysis, which was conducted independently for three times (Fig. 5l). Therefore, differentially accessible peaks exist in the *SVIL*, *FRYL* and *SHANK3* promoters.

Figure R23. Differentially accessible peaks were identified in the *SVIL*, *FRYL* and *SHANK3* promoters

Comment 5: Related to point #4 above, the image in Figure 5i is very difficult to discern. The Y-

axis for the ATAC-seq data is not labeled to show the read count range used and to further ensure that the range is consistent between treatment groups. It is also difficult to note the quality of the ATAC-seq data at such a narrow window. The authors should present the ATAC-seq data using a much larger window (~25-50kb) and include the location of genes. The authors should further denote in the figure which ATAC-seq signals/genomic regions were identified as significant peaks/sites of open chromatin and which were differentially accessible between treatment groups. I would suspect that promoters with significant increases in H3K9me3 to exhibit significant differential chromatin accessibility.

Response to comment 5: Thanks for your constructive comments. We apologize for the poor image in Figure 5i. We present the ATAC-seq data using a much larger window (29kb) and include the location of genes, we also have labeled the Y-axis now and ensure that the coverage is consistent between treatment groups (Fig. R24 becomes Figure 5k in the revised manuscript).

Figure R24. Differentially accessible peaks were identified in the *SVIL*, *FRYL* and *SHANK3* promoters

Comment 6: In regard to point #5 above, the authors should better explain how the H3K9me3 RT-qPCR results relate to the ATAC-seq image above. What do the colors (purple, gray and black) mean? Why were specific regions tested and not others (i.e. 3 regions for *SVIL* but only one for *FRYL* and *SHANK3*)?

Response to comment 6: Thanks for your insightful questions. In the promoters of *SVIL*, *FRYL* and *SHANK3*, we noticed several ATAC-seq peaks were “closed” in the serine-treatment group. Because H3K9me3 is a suppressive marker that causes chromatin inaccessibility, we examined the binding of H3K9me3 to those regions by using ChIP-qPCR. Different regions of primers used for ChIP-qPCR at the promoters are in different colors (Purple, gray and black). We changed the color into purple, gray and red for the third pair of primers for ChIP-qPCR (Fig. R25 becomes Fig. 5k).

We added the range of the position of those colors in Fig. 5k and included the detailed description for those colors in figure legends in the revised manuscript.

Figure R25. The different regions of primers used for ChIP-qPCR in the *SVIL*, *FRYL* and *SHANK3* promoters

Comment 7: The authors note that *RUNX1* was repressed via RNA-seq but not ATAC-seq. Did the authors focus solely on the *RUNX1* promoter region? Differentially expressed genes sometimes do not exhibit changes in promoter ATAC-seq signal. Were there any ATAC-seq peaks near *RUNX1* that exhibited differential accessibility? Is there any evidence for 3D looping of these peaks to the *RUNX1* promoter using publicly available HiC/HiChIP data? I think it would be worthwhile to investigate this given the importance of *RUNX1*.

Response to comment 7: Thanks for your insightful questions. We did not find differential accessibility peaks near *RUNX1* within 265kb (Fig. R26). We then performed literature review to assess whether there exists 3D looping in *RUNX1*. Surprisingly, several studies earlier suggested that enhancer–promoter loops are formed between the *Runx1* P1 promoter and neighboring regions and influence the chromatin states and chromatin-binding regulation⁴. As such, we envision that the formation of such loops may interfere with the ATAC-seq analysis.

Figure R26. There are no differential accessible peaks were identified in *RUNX1* in MKs treated with vehicle or serine

Comment 8: Does serine treatment lead to decreased SVIL protein levels?

Response to comment 8: Thanks for your insightful question. We explored potential regulation of SVIL by serine from two aspects. On one hand, we examined the protein level of SVIL in cells with serine treatment with WB using the megakaryocytic differentiation system *in vitro*. Consistent with the decreased mRNA level of *SVIL* caused by serine, the protein level of SVIL was also reduced by excessive serine (Fig. R27a becomes supplementary Fig. 6j). On the other hand, the level of SVIL was also reduced in CD41⁺ cells of 5TGM1 mice fed with control diet when compared with the serine-free diet as shown by immunofluorescence (Fig. R27a becomes supplementary Fig. 6k).

Figure R27. SVIL is decreased in MKs treated with serine at the protein level

a Western blot analysis of SVIL in differentiated cells at day 12 treated with vehicle or serine. **b** Immunofluorescence analysis for SVIL staining in MKs using CD41-FITC antibody (Green), SVIL antibody (red) and DAPI (Blue) in the BM of 5TGM1 mice received control diet and serine-free diet.

Comment 9: Similar to point #8 above, although the authors show that plasma cells express the highest levels of PHGDH (Fig 6B), the authors should demonstrate that this holds true at the protein level.

Response to comment 9: Thanks for your constructive suggestions. There are two cell types in BM including hematopoietic cells, which are differentiated from HSCs, and non-hematopoietic cells, among which the major cell type is BMSCs (Fig. R28a). To explore the cell origin of serine, we first isolated hematopoietic cellular components including progenitors (lin⁻c-Kit⁺ cells, LK), lymphocytes (B cells, B; plasma cells, P and T cells, T), myeloid cells (erythrocytes, Ery) as well as non-hematopoietic cells (BMSCs) based on their surface markers (Fig. R28b becomes Fig. 6a). We then assessed the levels of *Phgdh* mRNA and protein in LK, B, plasma cells, T, erythrocytes and BMSCs by using RT-qPCR and immunofluorescence. The results showed that both mRNA (Fig.

R28c becomes Fig. 6b) and protein levels of Phgdh were higher in plasma cells (MM cells) than in other cells (Fig. R28d becomes Fig. 6c and Supplementary Fig. 7a). Therefore, we conclude that plasma cells express the highest levels of Phgdh at not only the mRNA level but also the protein level.

Figure R28. The protein level of Phgdh is highly expressed in plasma cells in 5TGM1 mice

a Hematopoietic cellular components and non-hematopoietic cellular components in BM. **b** The gate strategy of LK, B, P, T, Ery and BMSCs with flow cytometry. **c** The mRNA expression of *Phgdh* and *Psph* in LK cells, B, plasma cells (P), T, erythrocytes (Ery) and BMSCs (gating in CD34⁺CD45⁻ cells) derived from 5TGM1 mice, BMSCs was used as control group. **d** The fluorescence image (left) and fluorescence intensities (right) of Phgdh in LK cells, B, P, T, Ery and BMSCs derived from 5TGM1 mice. Results represent means \pm SD; * $p < 0.05$, ** $p < 0.01$, *** $p < 0.001$, ns: not significant ($p > 0.05$). One-way ANOVA followed by multiple comparisons were used for **c**, **d**.

Comment 10: The authors have shown how MM cells inhibit megakaryopoiesis and thrombopoiesis. However, they also demonstrate that TPO suppresses MM progression and improves survival in MM mice. Regarding these experiments, can the authors speculate how TPO reduces serum serine levels? Can the authors further speculate how increased megakaryopoiesis and thrombopoiesis inhibits MM progression?

Response to comment 10: Thanks for your insightful comments. Because the serine level is reduced in 5TGM1 mice after TPO intervention, it is possible that TPO delays MM progression. It

is well known that there are two major sources of serine *in vivo* – from the diet or synthesized by cells. In previous studies, we found that serine level is elevated in 5TGM1 mice with MM progression, suggesting that serine can be used as an indicator for disease progression in 5TGM1 mice. on the other hand, the evidence that serine increases *in vivo* at the late stage of MM suggests an important role of endogenous serine synthesis. The mice in the TPO intervention group showed less tumor burden, decreased serine release from MM cells, and lower serine level *in vivo*.

As for how increased megakaryopoiesis/thrombopoiesis inhibits MM progression, we envision the following possibilities: First, PF4 is a major chemokine secreted from megakaryocyte⁵ and exhibits inhibitory effect on multiple myeloma⁶, suggesting that megakaryocytes may inhibit the progression of MM via PF4 secretion and/or other undefined factors. Second, several studies have reported that megakaryocytes maintain HSC homeostasis⁷ and are also involved in immune regulation⁸⁻¹⁰. Together, we speculate that megakaryocytes may inhibit the progression of MM by secreting anti-tumor factors such as PF4 or supporting normal hematopoiesis. In addition, megakaryocytes may function as immune cells that attenuate tumor progression. In the revised manuscript, we have included these potential mechanisms in the “Discussion” section.

Minor points:

comment 11: The authors need to be mindful of how they name genes when referring to human genes vs murine genes. Moreover, gene names need to be italicized to differentiate them from protein names.

Response to comment 11: Thank you very much for your critical comments. We have corrected the gene names in our revised manuscript and figures per your suggestion.

comment 12: On line 28 should thrombocytopenia be changed to thrombopoiesis?

Response to comment 12: We have changed thrombocytopenia to thrombopoiesis on line 28 (line 29) in revised manuscript.

comment 13: The authors need to define GSH on line 70.

Response to comment 13: We have defined glutathione as GSH on line 70 (line 71) in the revised manuscript.

comment 14: On lines 153-154, the authors state “the number of MKs was significantly lower in 5TGM1 mice with normal platelet counts than those with low platelet counts.” Is this correct? Should it not be the opposite?

Response to comment 14: Thank you very much for the comment. We apologize for the wrong description. We have changed the sentence from “the number of MKs was significantly lower in 5TGM1 mice with normal platelet counts than those with low platelet counts” to “the number of MKs was significantly lower in 5TGM1 mice with low platelet counts than those with normal platelet counts” on line 153-154 (line 159-160) in the revised manuscript.

comment 15: On line 162, “is” should be deleted.

Response to comment 15: We have deleted “is” on line 162 (line 168) in the revised manuscript.

comment 16: In Figure 1f, was there a significant difference in the number of MK cells in the BM between MM cells with low vs a normal number of platelets?

Response to comment 16: The number of MK cells in the BM is lower in MM patients with a low number of platelets than those with a normal number of platelets. We have added the statistical analysis in Figure 1f, which is replaced by Figure S1e in the revised manuscript.

comment 17: What is the meaning of m+0, m+1, m+2, m+3 in Figure 5a?

Response to comment 17: m+0, m+1, m+2, m+3 denote the number of ^{13}C labeled carbons derived from $^{13}\text{C}_3$ serine in each downstream metabolites. We have included the related explanations in the figure legends.

comment 18: In Fig 7a, vehicle is misspelled.

Response to comment 18: We corrected the spelling in Fig 7a in the revised manuscript.

comment 19: Line 434, a space is required between “including” and “RUNX1”.

Response to comment 19: We have added a space between “including” and “RUNX1” in the revised manuscript (line 487).

References:

1. Miao, W., *et al.* Integrative ATAC-seq and RNA-seq Analysis of the Longissimus Muscle of Luchuan and Duroc Pigs. *Front Nutr* **8**, 742672 (2021).
2. Mao, X.Q., *et al.* RNA-seq and ATAC-seq analyses of multilineage differentiating stress enduring cells: Comparison with dermal fibroblasts. *Cell Biol Int* **46**, 1480-1494 (2022).
3. Yu, S., *et al.* BMP2-dependent gene regulatory network analysis reveals Klf4 as a novel transcription factor of osteoblast differentiation. *Cell Death Dis* **12**, 197 (2021).
4. Li, C.C., *et al.* Pre-configuring chromatin architecture with histone modifications guides hematopoietic stem cell formation in mouse embryos. *Nat Commun* **13**, 346 (2022).
5. Norozi, F., Shahrabi, S., Hajizamani, S. & Saki, N. Regulatory role of Megakaryocytes on Hematopoietic Stem Cells Quiescence by CXCL4/PF4 in Bone Marrow Niche. *Leuk Res* **48**, 107-112 (2016).
6. Liang, P., *et al.* Platelet factor 4 induces cell apoptosis by inhibition of STAT3 via up-regulation of SOCS3 expression in multiple myeloma. *Haematologica* **98**, 288-295 (2013).
7. Bruns, I., *et al.* Megakaryocytes regulate hematopoietic stem cell quiescence through CXCL4 secretion. *Nat Med* **20**, 1315-1320 (2014).
8. Campbell, R.A., *et al.* Human megakaryocytes possess intrinsic antiviral immunity through regulated induction of IFITM3. *Blood* **133**, 2013-2026 (2019).
9. Zufferey, A., *et al.* Mature murine megakaryocytes present antigen-MHC class I molecules to T cells and transfer them to platelets. *Blood Adv* **1**, 1773-1785 (2017).
10. Liu, C., *et al.* Characterization of Cellular Heterogeneity and an Immune Subpopulation of Human Megakaryocytes. *Adv Sci (Weinh)* **8**, e2100921 (2021).

Reviewers' Comments:

Reviewer #1:

Remarks to the Author:

My comments have been sufficiently answered.

Reviewer #2:

Remarks to the Author:

The authors answered all my concerns.

Reviewer #3:

Remarks to the Author:

The authors have addressed most of my concerns. Congrats to the authors!

Reviewer #4:

Remarks to the Author:

The authors have addressed all my concerns.